# Proactive distractor suppression in early visual cortex

David Richter[1,2,3]*, Dirk van Moorselaar[1,2], Jan Theeuwes[1,2,4]

[1]Vrije Universiteit Amsterdam, Amsterdam, Netherlands; [2]Institute Brain and Behavior Amsterdam (iBBA), Amsterdam, Netherlands; [3]Mind, Brain and Behavior Research Center (CIMCYC), University of Granada, Granada, Spain; [4]William James Center for Research, ISPA-Instituto Universitario, Lisbon, Portugal

## eLife Assessment

This **important** and well-written study uses functional neuroimaging in human observers to provide **compelling** evidence that activity in the early visual cortex is suppressed at locations that are frequently occupied by a task-irrelevant but salient item. This suppression appears to be general to any kind of stimulus and also occurs in advance of any item actually appearing. The work will be of great interest to psychologists and neuroscientists examining attention, perception, learning and prediction.

*For correspondence:
david.richter.work@gmail.com

Competing interest: The authors declare that no competing interests exist.

**Abstract** Avoiding distraction by salient yet irrelevant stimuli is critical when accomplishing daily tasks. One possible mechanism to accomplish this is by suppressing stimuli that may be distracting such that they no longer compete for attention. While the behavioral benefits of distractor suppression are well established, its neural underpinnings are not yet fully understood. In a functional MRI (fMRI) study, we examined whether and how sensory responses in early visual areas show signs of distractor suppression after incidental learning of spatial statistical regularities. Participants were exposed to an additional singleton task where, unbeknownst to them, one location more frequently contained a salient distractor. We analyzed whether visual responses in terms of fMRI BOLD were modulated by this distractor predictability. Our findings indicate that implicit spatial priors shape sensory processing even at the earliest stages of cortical visual processing, evident in early visual cortex as a suppression of stimuli at locations which frequently contained distracting information. Notably, while this suppression was spatially (receptive field) specific, it did extend to nearby neutral locations and occurred regardless of whether distractors, nontarget items, or targets were presented at this location, suggesting that suppression arises before stimulus identification. Crucially, we observed similar spatially specific neural suppression even if search was only anticipated, but no search display was presented. Our results highlight proactive modulations in early visual cortex, where potential distractions are suppressed preemptively, before stimulus onset, based on learned expectations. Combined, our study underscores how the brain leverages implicitly learned prior knowledge to optimize sensory processing and attention allocation.

## Introduction

Selective attention is pivotal in our daily interactions with the environment, enabling us to navigate complex settings and perform challenging tasks. Essential for this capability is the suppression of nonessential stimuli allowing us to maintain focus on critical activities, like driving a car, by filtering out salient yet irrelevant information that could otherwise divert our attention. The behavioral advantages of distractor suppression have been extensively documented, underscoring its significance

in enhancing cognitive efficiency and task performance (*Chelazzi et al., 2019*; *Luck et al., 2021*; *Theeuwes et al., 2022*; *van Moorselaar and Slagter, 2020*). However, the neural foundations of distractor suppression remain an area of active inquiry. Traditional models of attention posit that the visual system is adept at prioritizing relevant stimuli while suppressing irrelevant ones, a process facilitated by both bottom-up (stimulus-driven) and top-down (goal-directed) mechanisms (*Awh et al., 2012*). Yet, recent advancements suggest a more nuanced understanding, incorporating the role of prior knowledge derived by statistical learning (*Theeuwes et al., 2022*). Indeed, salient distractors result in less behavioral interference if they appear at locations with higher probability, showing that implicitly learned prior knowledge can aid distractor suppression (*Ferrante et al., 2018*; *Wang and Theeuwes, 2018a*; *Wang and Theeuwes, 2018b*). Here, we approach distractor suppression through this lens of incidental statistical learning and investigate its effect on visual processing to gain new insights into the neural mechanisms underpinning attentional selection.

The visual system shows a remarkable sensitivity to statistical regularity, including suppressed neural responses to predicted compared to surprising stimuli through the visual hierarchy (*Melloni et al., 2012*; *Richter et al., 2018*; *Richter and de Lange, 2019*; *Won et al., 2020*). Therefore, the visual system is likely to play a role in facilitating distractor suppression using acquired prior knowledge, attenuating neural responses to predictable distractors possibly already at the first stages of the visual cortical hierarchy, in early visual cortex (EVC; primary and second visual cortex). Indeed, in line with this proposition, recent functional MRI (fMRI) work suggests that statistical regularities during visual search modulate neural processing in EVC (*Adam and Serences, 2021*; *Beffara et al., 2023*; *Won et al., 2020*; *Zhang et al., 2022*). Specifically, BOLD responses, including in EVC, are enhanced when targets (*Beffara et al., 2023*) or suppressed when distractors (*Zhang et al., 2022*) appear at predictable locations. Together these findings are consistent with the notion of spatial priority maps, reflecting prior knowledge, facilitating visual search by both target enhancement and distractor suppression (*Awh et al., 2012*; *Bisley and Goldberg, 2010*; *Fecteau and Munoz, 2006*; *Itti and Koch, 2000*; *Serences and Yantis, 2006*; *Theeuwes et al., 2022*).

However, multiple questions remain on how implicit prior knowledge of distractor probabilities, following statistical learning, shapes neural responses in EVC. It is at present unknown how spatially specific distractor suppression is, as previous studies using fMRI utilized paradigms in which entire regions were likely to contain distractors (*Zhang et al., 2022*). This leaves open the question whether learned suppression in EVC, which at the behavioral level is often characterized by a gradient around the high-probability distractor location (HPDL) (*Wang and Theeuwes, 2018a*), is spatially specific or alternatively more widespread. Elucidating the spatial extent of suppression yields valuable insights into the neural mechanism underlying suppression. Moreover, it is unclear whether suppression in EVC is stimulus-specific, affecting only distractors, or stimulus-unspecific as proposed by a recent study (*Zhang et al., 2022*). Finally, it is widely contested whether spatial priority maps underlying distractor suppression are deployed proactively (i.e. predictively before stimulus onset), as suggested by some accounts (*Huang et al., 2021*; *Huang et al., 2022*), or reactively following visual search onset (*Chang et al., 2023*; *Moher and Egeth, 2012*).

Here, we addressed these questions by exposing human volunteers to a variant of the additional singleton task (*Theeuwes, 1991*; *Theeuwes, 1992*) with omission trials while recording fMRI. The task consisted of a search display with target and distractor stimuli. Crucially, salient distractor stimuli appeared more often in one location, allowing participants to learn these contingencies resulting in facilitated visual search performance. To foreshadow the results, data show that EVC is sensitive to spatial statistical regularities, evident as a suppression of neural responses corresponding to locations that frequently contained distracting stimuli. However, this suppression was surprisingly broad, encompassing nearby neutral locations. Critically, we also demonstrate that this suppression emerges proactively, in conditions in which the search display is expected but not actually presented (omission trials). Combined our data shed new light on the mechanisms underlying distractor suppression and the interaction between attentional control and perception.

## Results

Participants performed a variant of the additional singleton task (*Theeuwes, 1991*; *Theeuwes, 1992*), as illustrated in *Figure 1A*. During search trials, participants were presented with eight bar stimuli arranged in a circular configuration around the central fixation dot. Seven of the eight stimuli were

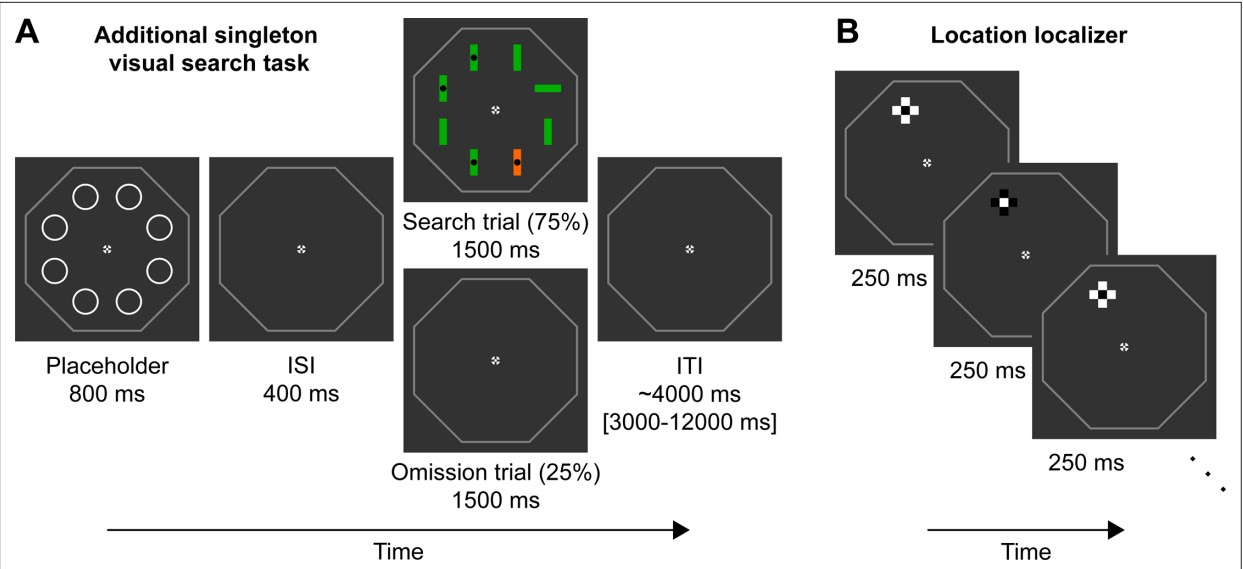

**Figure 1.** Paradigm. (**A**) Example trial of the additional singleton visual search task. Trials started with a placeholder display (800 ms duration) signifying trial onset, followed by a fixed interstimulus interval (ISI) of 400 ms. On 75% of trials a search trial appeared next for 1500 ms. On 25% of trials, an omission trial (identical to the ISI display) was shown instead. Trials ended with a variable intertrial interval (ITI) of ~4000 ms. In the example search trial, the target is the horizontal green bar in the right upper corner, because it has a different orientation compared to the other seven bar stimuli. Because the target contains no black dot in the center, the correct response is 'no'. The highly salient distractor is the orange vertical bar. The central fixation cross and octagonal outline surrounding the possible search area were presented throughout the experiment. (**B**) Three example color inversions of the location localizer, sampling the upper left location. The location localizer cross had the same size and location as the two possible bar stimuli (horizontal and vertical) during the search trial overlaid over each other, thus sampling neural populations responsive to this location. The cross flickered at 4 Hz for 12 s for each location of interest, sampling each location 8 times per run.

of the same orientation (vertical or horizontal) and participants were tasked to report whether or not the bar stimulus with a unique orientation (e.g. the horizontal bar among the vertical orientations in the example trial in *Figure 1A*) contained a black circle at its center. On most trials one of the stimuli had a unique color (orange if all other stimuli were green or vice versa) rendering it a color singleton distractor. Only four locations contained distractors or targets. Specifically, each location of interest was followed by a noninterest location in a clockwise direction (i.e. alternating locations of interest and no interest), creating an evenly spaced pattern of interest and noninterest sites. Unbeknownst to participants, while targets appeared with equal probability across the four locations, the color distractor appeared four times more often in one location relative to the other locations (counterbalanced across participants). Participants could exploit this statistical regularity by learning to suppress distractors presented at this HPDL. Trial onset was cued with an otherwise uninformative placeholder display (eight white circles), signaling imminent search trial onset. On 25% of trials, no search display was presented following the placeholder, representing an omission trial.

## Behavioral facilitation by distractor suppression

First, we evaluated whether participants successfully learned and exploited the underlying distractor contingencies to facilitate behavior. During distractor present trials, we evaluated RTs contingent on whether the silent distractor was shown at the HPDL, a neutral location nearby the HPDL (NL-near), or the neutral location furthest away from the HPDL (NL-far). As shown in *Figure 2A*, RTs were significantly affected by distractor location (main effect of distractor location on RT: $F_{(3,81)} = 9.27$, p<0.001, $\eta_p^2 = 0.26$). RTs were fastest when no distractor was present (833 ms; post hoc tests: Absent vs NL-far: $t_{(27)} = 4.27$, $p_{holm}<0.001$, $d=0.31$; Absent vs NL-near: $t_{(27)} = 4.49$, $p_{holm}<0.001$, $d=0.33$; Absent vs HPDL $t_{(27)} = 1.70$, $p_{holm} = 0.188$, $d=0.12$; $BF_{10}=1.90$). For trials containing a distractor, the fastest RTs were observed for distractor stimuli appearing at the HPDL (848 ms; HPDL vs NL-far: $t_{(27)} = 2.57$, $p_{holm} = 0.036$, $d=0.19$; HPDL vs NL-near: $t_{(27)} = 2.79$, $p_{holm} = 0.026$, $d=0.20$), showing clear evidence of behavioral facilitation due to learning of the statistical regularities governing distractor appearances. Responses were similarly slow at the near neutral (871 ms) and far neutral locations (870 ms; NL-far vs NL-near: $t_{(27)} = $

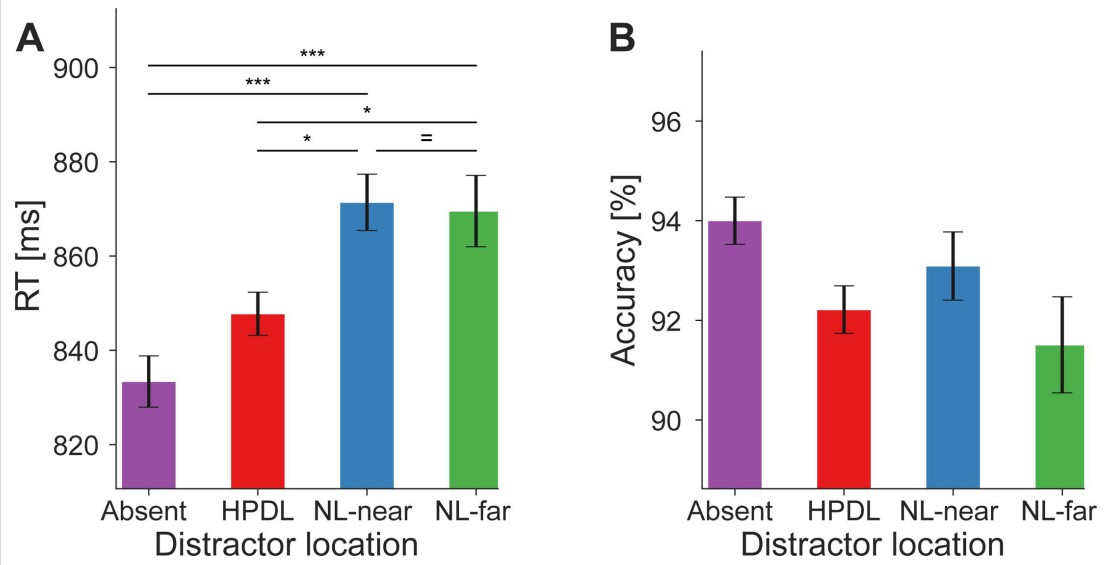

**Figure 2.** Behavioral facilitation by distractor suppression. (**A**) Reaction times (RTs in ms; ordinate) were faster when distractor stimuli appeared at the high-probability distractor location (HPDL) compared to neutral location nearby the HPDL (NL-near) or neutral location furthest away from the HPDL (NL-far) locations. RTs were fastest when no distractor was present. (**B**) Response accuracy (in percent; ordinate) for each distractor location. Error bars denote within-subject SEM. Asterisks indicate statistically significant post hoc tests: *p<0.05, ***p<0.001, = BF$_{10}$<1/3.

The online version of this article includes the following source data for figure 2:

**Source data 1.** Behavioral facilitation by distractor suppression – reaction time (RT) data.

**Source data 2.** Behavioral facilitation by distractor suppression – accuracy data.

0.22, $p_{holm}$ = 0.828, d=0.02; BF$_{10}$=0.20), highlighting the spatial specificity of learned suppression. No reliable differences in response accuracy between the distractor locations were observed (***Figure 2B***); main effect of distractor location on accuracy: $F_{(1.84,49.70)}$=2.52, p=0.095, $\eta_p^2$ = 0.85.

### Distractor suppression in EVC

Given that the behavioral results showed evidence of suppression, we next assessed neural modulations in EVC. Analyses were performed using a region of interest (ROI)-based approach. In this analysis, for each participant, we first determined the neural populations in EVC with receptive fields corresponding to the four stimulus locations of interest using independent location localizer data (***Figures 1B and 3A***, left). Next, these four ROI masks were applied to the search or omission trial data, thus yielding for each trial and ROI mask a contrast parameter estimate reflecting the activation elicited by the stimulus presented at the specific location. See ***Figure 3A*** (right) and the associated figure text for an example trial. Each trial thereby supplied four data points to the results figure (four locations, three stimulus types). Because the analysis depended on differentiation of neural populations responding to stimuli at the four locations of interest, analyses were constrained to EVC, which allowed for reliable mask definition. Moreover, to better separate neural responses between locations, each location of interest was separated from the next by a location of no interest, which only served as a filler to make visual search more challenging. This analysis procedure assessed two distinct factors of BOLD suppression, with possible outcomes illustrated in ***Figure 3B***. First, we tested whether BOLD suppression in EVC was spatially specific (left panel vs middle and right panels in ***Figure 3B***). Second, we assessed whether suppression was stimulus-specific (left, middle vs right panel).

Results, depicted in ***Figure 4A***, showed that BOLD responses in EVC differed in magnitude between stimulus types (main effect of stimulus type: $F_{(2,54)}$ = 35.85, p<0.001, $\eta_p^2$ = 0.57) with targets on average eliciting larger BOLD responses compared to distractors ($t_{(27)}$ = 3.78, $p_{holm}$<0.001, d=0.08) and neutral stimuli ($t_{(27)}$ = 8.45, $p_{holm}$<0.001, d=0.17). In addition, distractors evoked larger responses than neutral stimuli ($t_{(27)}$ = 4.67, $p_{holm}$<0.001, d=0.10). These results likely reflect a top-down modulation due to target relevance and a bottom-up effect of distractor salience.

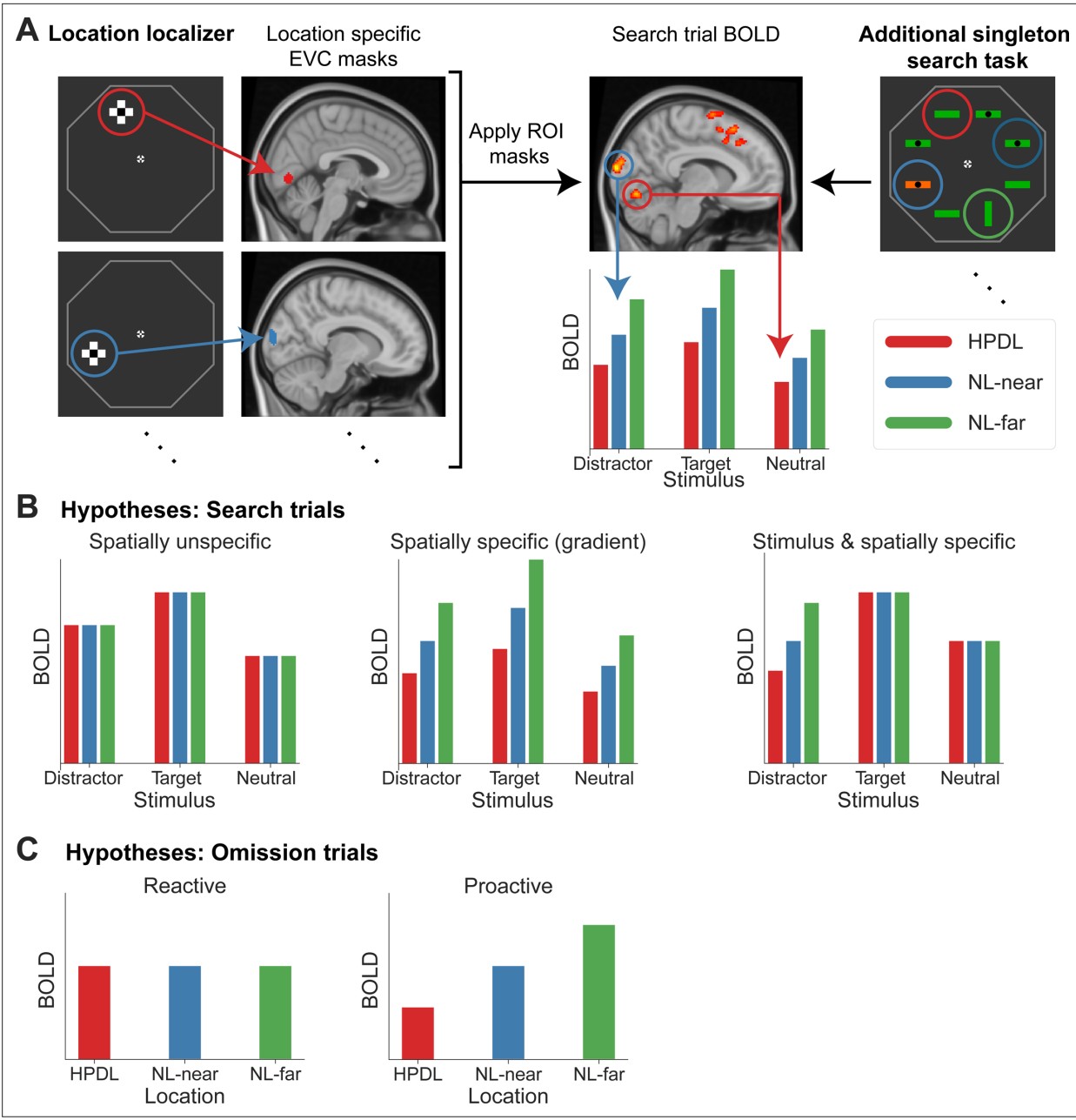

**Figure 3.** Illustration of the analysis rationale and hypotheses. (**A**) Region of interest (ROI) analysis procedure. During an independent location localizer task (left) checkerboard cross patterns (flickering black and white at 4 Hz) were presented at the locations corresponding to the bar locations during the search task (right). Using BOLD activations in early visual cortex (EVC) from this localizer, location-specific ROI masks were created for the four locations of interest (high-probability distractor location [HPDL], two neutral location nearby the HPDL [NL-near], neutral location furthest away from the HPDL [NL-far]). The masks were then applied to the additional singleton task and the contrast parameter estimates (BOLD) during search trials extracted in a stimulus and location-specific manner. To illustrate, in the example above we assume that the HPDL was at the upper left location (red circle; determined by the statistical regularities throughout the search task). The example search trial contained a neutral stimulus at the HPDL (red circle), a salient distractor at the left NL-near (blue circle), a neutral stimulus at the right NL-near (dark blue circle), and a target stimulus at the NL-far (green circle) location. Therefore, the data provided by this trial was a neutral stimulus at HPDL (red arrow), a distractor at NL-near (blue arrow), a neutral stimulus at the other NL-near (not depicted), and a target at NL-far (not depicted). Therefore, each trial provided multiple location-specific data points. Specifically, data for each stimulus type and location combination were first estimated across trials and then extracted using the ROI-based approach. Data across the two NL-near locations were averaged. For further details, see *Materials and methods: Statistical analysis* and *ROI definition*. The same ROI analysis was performed for omission trials, except that by design, omission trials did not contain stimuli, hence resulting in only location-specific activation data points. (**B**) Potential outcomes for search trials. We distinguish between two factors modulating BOLD responses during search trials. First, we asked whether modulations in EVC are spatially specific. Illustrated on the left is a spatially unspecific modulation, affecting neural populations

*Figure 3 continued on next page*

*Figure 3 continued*

with receptive fields at all three locations equally. The middle panel depicts a spatially specific modulation with a gradient of increasing suppression the closer a location is to the HPDL. Second, we ask whether BOLD modulations are stimulus-specific, that is selectively suppressing only distractor stimuli, but not target and neutral stimuli (right panel). (**C**) Additionally, we distinguished between reactive compared to proactive spatial modulations by contrasting BOLD during omission trials. Reactive modulations (i.e. following search display onset) result in no spatially specific effects during omission trials (left panel), because no search display was shown. In contrast, proactive suppression yields spatially specific BOLD modulations during omission trials due to the deployment of spatial priority maps by anticipated search (right panel).

With respect to the key manipulation of distractor predictability, results revealed a distinct pattern of location-specific BOLD suppression (main effect of location: $F_{(2,54)}$ = 7.33, p=0.002, $\eta_p^2$ = 0.21). Neural populations with receptive fields corresponding to the HPDL showed significantly reduced BOLD responses compared to the diagonally opposite neutral location (NL-far; post hoc test HPDL vs NL-far: $t_{(27)}$ = 3.53, $p_{holm}$ = 0.003, d=0.59). Intriguingly, and counter to the observed behavior, this suppression was not confined to the HPDL but also extended to close by neutral locations (NL-near vs NL-far: $t_{(27)}$ = 3.06, $p_{holm}$ = 0.007, d=0.51). BOLD responses between HPDL and NL-near locations

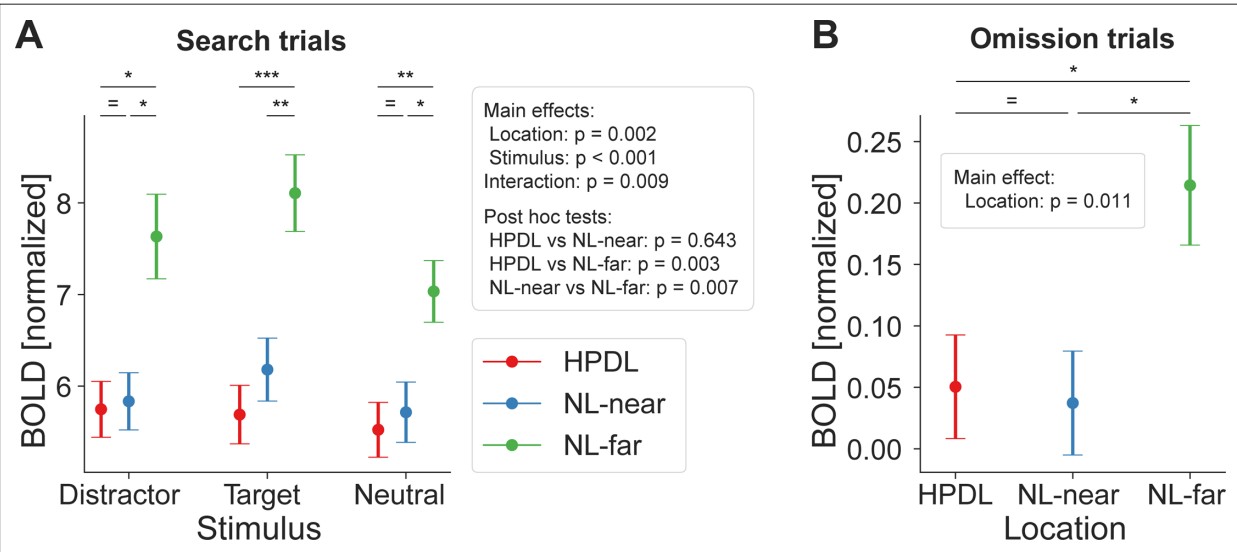

**Figure 4.** Distractor suppression in early visual cortex. (**A**) fMRI BOLD responses (ordinate) during search trials, split into stimulus types (abscissa). Color denotes locations based on distractor contingencies with red = high-probability distractor location (HPDL), blue = neutral locations near the HPDL (NL-near), green = neutral locations far from the HPDL (NL-far; diagonally opposite from the HPDL). BOLD responses were systematically suppressed for all stimuli occurring at the HPDL and NL-near compared to NL-far. (**B**) Corresponding results for omission trials. Neural populations with receptive fields at the HPDL and NL-near locations were suppressed compared to those with receptive fields at the NL-far location. Note that BOLD responses are overall close to zero, which is expected given that no display was shown at this time. Error bars denote within-subject SEM. Asterisks indicate statistically significant pairwise comparisons within stimulus types: *p<0.05, **p<0.01, ***p<0.001, = BF₁₀<1/3.

The online version of this article includes the following source data and figure supplement(s) for figure 4:

**Source data 1.** Distractor suppression in early visual cortex – search trials.

**Source data 2.** Distractor suppression in early visual cortex – omission trials.

**Figure supplement 1.** fMRI results generalize across region of interest (ROI) mask sizes.

**Figure supplement 1—source data 1.** fMRI results generalize across region of interest (ROI) mask sizes – search trials.

**Figure supplement 1—source data 2.** fMRI results generalize across region of interest (ROI) mask sizes – omission trials.

**Figure supplement 2.** fMRI results did not depend on location-specific normalization of BOLD responses.

**Figure supplement 2—source data 1.** fMRI results did not depend on location-specific normalization of BOLD responses – search trials.

**Figure supplement 2—source data 2.** fMRI results did not depend on location-specific normalization of BOLD responses – omission trials.

**Figure supplement 3.** Priming does not explain distractor suppression.

**Figure supplement 3—source data 1.** Priming does not explain distractor suppression.

**Figure supplement 4.** Region of interest (ROI) location masks generalize across localizer runs.

**Figure supplement 4—source data 1.** Region of interest (ROI) location masks generalize across localizer runs.

did not reliably differ (HPDL vs NL-near: $t_{(27)}$ = 0.47, $p_{holm}$ = 0.643, $d$=0.08; $BF_{10}$=0.19). This pattern of results was present regardless whether the distractor, target, or a neutral stimulus presented at the HPDL and NL-near locations compared to NL-far (6 of 6 paired t-tests: p<0.05; see table in ***Supplementary file 1***). In sum, these results show that neural responses in EVC were significantly modulated by the distractor contingencies, evident as reduced BOLD responses in neural populations with receptive fields at the HPDL and neutral locations near the location of the frequent distractor (NL-near), relative to the neutral location diagonally across the HPDL (NL-far). However, surprisingly neural suppression in EVC had a less spatially selective (i.e. broader) pattern of suppression compared to the behavioral facilitation, which was specific to the HPDL (***Figure 2***).

## Proactive neural suppression during omissions

Given the location-specific suppression seen above, we then tested whether neural suppression was applied proactively or reactively by analyzing omission trials, during which only an uninformative placeholder, but no search display, was presented. Proactive modulations ought to result in spatially specific modulation as illustrated in ***Figure 3C*** (right panel), while reactive deployments of spatial priority maps should result in no location-specific modulations (***Figure 3C***, left panel). Results during omission trials, depicted in ***Figure 4B***, showed a similar suppression pattern as during search trials at the HPDL and NL-near (main effect of location: $F_{(2,54)}$ = 4.93, p=0.011, $\eta_p^2$ = 0.15). Crucially, BOLD responses corresponding to the HPDL were suppressed compared to the NL-far location ($t_{(27)}$ = 2.61, $p_{holm}$ = 0.024, $d$=0.45), as well as for the NL-near location compared to NL-far ($t_{(27)}$ = 2.82, $p_{holm}$ = 0.020, $d$=0.49). Neural responses at HPDL and NL-near did not reliably differ ($t_{(27)}$ = 0.21, $p_{holm}$ = 0.835, $d$=0.04; $BF_{10}$=0.21). This finding suggests a proactive, or predictive, nature of distractor suppression in visual cortex, activated merely by cues predicting the likely onset of a search display.

## Ruling out attentional strategies

It is possible that the observed suppression could be attributed to a strategy whereby participants chose to attend the far neutral location more – i.e., attending away from the HPDL, thereby causing the larger BOLD response at NL-far compared to NL-near and HPDL. To assess this possibility, we analyzed behavioral data in a target contingent analysis, because on this account faster RTs and higher accuracies for targets at NL-far would be expected, reflecting the prioritized processing of NL-far compared to the other locations. Results, depicted in ***Figure 5—figure supplement 1A***, showed that RTs did not differ for targets at the three locations (main effect of target location on RT: $F_{(2,54)}$ = 0.43, p=0.66, $\eta_p^2$ = 0.02). Responses for targets at NL-far (833 ms) were similar to targets at NL-near (822 ms; post hoc test NL-far vs NL-near: $t_{(27)}$ = 0.68, $p_{holm}$ = 1.0, $d$=0.09) and at the HPDL (836 ms; post hoc test NL-far vs HPDL: $t_{(27)}$ = 0.20, $p_{holm}$ = 1.0, $d$=0.03). Similar results were observed in terms of response accuracy (***Figure 5—figure supplement 1B***). Again, there was overall no effect of target location on accuracy (main effect: $F_{(1.53,39.84)}$=0.33, p=0.724, $\eta_p^2$ = 0.01), with targets at NL-far (93.9% accuracy) resulting in similar accuracies compared to targets at HPDL (94.8% accuracy; post hoc test NL-far vs HPDL: $t_{(27)}$ = 0.50, $p_{holm}$ = 1.0, $d$=0.12) and NL-near (post hoc test NL-far vs NL-near: 93.4% accuracy; $t_{(27)}$ = 0.30, $p_{holm}$ = 1.0, $d$=0.07). Indeed, Bayesian tests showed evidence for the absence of a difference in both accuracy (NL-far vs NL-near: $BF_{10}$=0.209, and NL-far vs HPDL: $BF_{10}$=0.231) and RTs (NL-far vs NL-near: $BF_{10}$=0.243, and NL-far vs HPDL: $BF_{10}$=0.207). Therefore, in terms of both RT and response accuracy, behavioral results suggest that it is unlikely that increased attention toward the NL-far location explains the observed neural suppression pattern.

Moreover, following MRI scanning, participants filled in a questionnaire probing their explicit knowledge of the distractor contingencies. At the group level, results indicated little knowledge of the statistical regularities with only 35.7% of participants identifying the correct HPDL. A chi-square goodness-of-fit test indicated no significant deviation from chance level of 25% ($\chi^2_{(1, N = 28)}$=1.71, p=0.190; given that target and distractor stimuli appeared only at four locations, chance is arguably 25%), thus indicating that our results are unlikely to be explained by explicit attention strategies. Further evidence supporting that our results cannot be explained by explicit attentional strategies is provided in ***Figure 5***, where we replicate our main fMRI analyses in a subsample of participants that indicated an incorrect HPDL in the questionnaire. Critically, results were highly similar to ***Figure 4*** in this subsample, providing strong evidence that distractor suppression in the present study does not reflect explicit attentional strategies. Instead, the combined evidence favors an account emphasizing

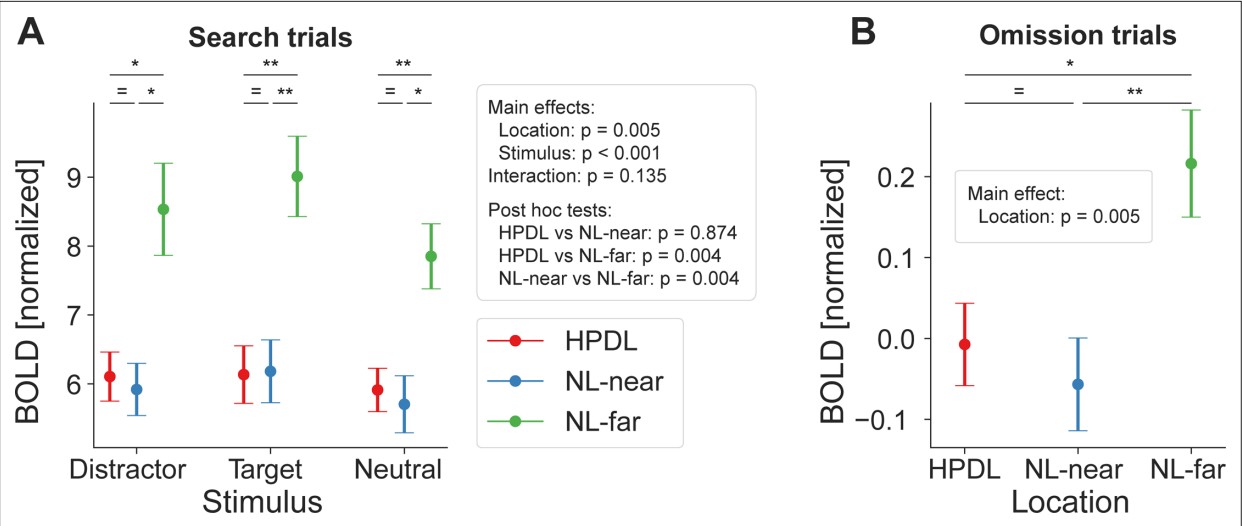

**Figure 5.** Ruling out explicit attentional strategies: distractor suppression in a subsample of participants with incorrect high-probability distractor location (HPDL) choices in the questionnaire. Results for search (**A**) and omission (**B**) trials were highly similar to the main results using the full sample. Significant suppression of BOLD responses at both the HPDL and neutral location nearby the HPDL (NL-near) locations compared to the neutral location furthest away from the HPDL (NL-far) location were evident during search and omission trials in the subsample with incorrect responses for the HPDL location on the questionnaire probing explicit knowledge of the distractor contingencies. Error bars denote within-subject SEM. Asterisks indicate statistically significant pairwise comparisons within stimulus types: *p<0.05, **p<0.01, = $BF_{10}<1/3$.

The online version of this article includes the following source data and figure supplement(s) for figure 5:

**Source data 1.** Distractor suppression in a subsample of participants with incorrect high-probability distractor location (HPDL) choices in the questionnaire – search trials.

**Source data 2.** Distractor suppression in a subsample of participants with incorrect high-probability distractor location (HPDL) choices in the questionnaire – omission trials.

**Figure supplement 1.** No behavioral prioritization of targets at neutral location furthest away from the HPDL (NL-far).

**Figure supplement 1—source data 1.** No behavioral prioritization of targets at neutral location furthest away from the HPDL (NL-far) – reaction time (RT) data.

**Figure supplement 1—source data 2.** No behavioral prioritization of targets at neutral location furthest away from the HPDL (NL-far) – accuracy data.

distractor contingencies learned by implicit visual statistical learning guiding subsequent visual search, and that this implicit guide is spatially broad in EVC (affecting NL-near and HPDL similarly), as well as proactively deployed (evident also during omission trials).

## Control analyses

We performed several control analyses to rule out alternative interpretations of our data. First, to ensure that our results did not depend on the a-prior selected, but arbitrary ROI mask size (20 voxels per location, i.e. 80 voxels total, or 640 mm³) we repeated all main analyses across a variety of mask sizes ranging from 10 to 500 voxels per location (*Figure 4—figure supplement 1*). Results were qualitatively similar across small and medium mask sizes, yielding significant simple main effects of location until very large mask sizes of 200 or more voxels per location, as expected by the predictable drop in location selectivity. Thus, our results were not dependent on the precise ROI mask size supporting the reliability of our results.

While the HPDL varied between participants, it remained constant within each individual throughout the experiment. This constancy opens the possibility that observed differences in BOLD signal strength could be attributed to intrinsic variations in BOLD sensitivity across different areas of the visual cortex, independent of the experimental manipulations (distractor contingencies). To minimize such potential confounds the results reported in *Figure 4* reflect BOLD responses normalized for each location using independent localizer data to correct for hemodynamic differences between locations (see Materials and methods for details). However, to ensure that our results did not depend on this normalization step, we repeated the analyses without normalization (*Figure 4—figure supplement 2*). Results were

qualitatively similar to the normalized results, suggesting that the normalization method did not introduce the observed results.

## Discussion

Here, we asked how distractor suppression aided by implicit spatial priors, derived from statistical learning, is implemented in EVC. To this end we exposed participants to visual search displays where salient distractors appeared more often at one location. Results showed that neural responses in populations responding to the HPDL and to nearby neutral locations were attenuated compared to responses in faraway neutral locations. Additionally, suppression was stimulus unspecific, arising for distractor, target, and neutral stimuli. Critically, we also showed that suppression in EVC arose even during omission trials in which no search display was presented, indicating that the mere anticipation of search was sufficient to instantiate neural suppression.

### Distractor suppression in EVC

In line with previous studies our data support that EVC is sensitive to statistical regularities (*Adam and Serences, 2021*; *Beffara et al., 2023*; *Melloni et al., 2012*; *Richter et al., 2018*; *Richter and de Lange, 2019*; *Won et al., 2020*; *Zhang et al., 2022*) highlighting that perception fundamentally relies on (implicit) priors (*de Lange et al., 2018*). Our results extend previous studies on target learning (*Beffara et al., 2023*) to distractor suppression and show that EVC responses were suppressed at the HPDL and nearby neutral locations. Suppression was stimulus-unspecific, resulting in attenuated EVC responses irrespective of the stimulus presented, complementing previous studies (*Wang et al., 2019*; *Zhang et al., 2022*). From the perspective of optimizing perception and task performance, this may seem counterintuitive as neural representations of target stimuli at or near the HPDL were also suppressed, even though targets occurred equiprobably at all locations. However, despite this suppression we found no behavioral costs for target identification at these locations.

Indeed, the discrepancy between the spatially broad neural suppression in EVC, extending beyond the HPDL to encompass neural populations coding for nearby neutral locations, and the more localized behavioral results, showing distractor suppression specific to the HPDL, further raises interesting questions about the relationship between neural activity in early visual areas and subsequent processing contributing to distractor suppression. Given the size of the stimulus display (radius of 5 degrees visual angle [DVA]), the small receptive field sizes in EVC (*Chen et al., 2009*; *De Valois et al., 1982*; *Dow et al., 1981*; *Dumoulin and Wandell, 2008*), and prior behavioral studies showing location-specific distractor suppression, and sometimes gradients of suppression, using smaller stimulus displays (*Huang et al., 2022*; *Wang and Theeuwes, 2018a*), make the broadness of the observed neural suppression surprising. Combined, these findings suggest that later processing stages in higher visual areas and downstream from visual cortex refine the initial neural suppression in EVC to facilitate task performance in a more nuanced fashion. It is plausible that areas in parietal and frontal cortex, including the dorsal attention network (*Beffara et al., 2023*; *Zhang et al., 2022*), contribute to this refinement. However, this does not mean that suppression in EVC merely reflects an epiphenomenon of the instantiation of spatial priority maps in later stages of cortical processing. Instead, EVC may form an initial spatial bias that prioritizes a broad area of the visual field. This initial spatial bias account seems well in line with the proactive nature of suppression discussed in more detail later. An open question is whether the broad suppression reflects limitations of the neural mechanisms underlying distractor suppression in EVC, such as less spatially specific feedback, or a neural efficiency trade-off, in which a more accurate suppression is not sufficiently beneficial to warrant the required metabolic demand of more precise modulations in EVC.

Overall, our results fit well within the framework of spatial priority maps (*Awh et al., 2012*; *Bisley and Goldberg, 2010*; *Fecteau and Munoz, 2006*; *Itti and Koch, 2000*; *Serences and Yantis, 2006*), de-emphasizing processing at and near the HPDL already at the earliest stages of the cortical visual hierarchy. Spatial priority maps are frequently proposed to reside at the top of a hierarchy of topographic maps, with feature-specific maps at the bottom of this hierarchy of topographic maps (*Fecteau and Munoz, 2006*; *Itti and Koch, 2000*; *Itti and Koch, 2001*; *Zelinsky and Bisley, 2015*). From this perspective, the lack of stimulus specificity arises because spatial priority maps are implemented at a higher level than the feature-specific maps. If priority signals from these later processing

stages are then fed back into early sensory processing, the resulting modulations are necessarily stimulus-unspecific, as seen here. Moreover, given the wider receptive fields at later visual processing stages and reduced spatial acuity due to less spatially specific feedback signals (*Gilbert and Wiesel, 1989*; *Nassi and Callaway, 2009*), the broad spatial suppression observed in EVC fits well into this framework.

## Proactive distractor suppression

As argued above, we believe that through statistical learning the weights within a spatial priority map are adjusted such that the location that more frequently contains a distractor is suppressed (*Theeuwes et al., 2022*). Critically, the current findings indicate that this suppression is instantiated before stimulus identification is completed, suggesting that distractor suppression arises proactively. Indeed, our results from omission trials strongly support such an interpretation. Even though during omission trials no search display was presented, the same spatially specific suppression at the HPDL and nearby locations was found. Thus, the mere anticipation of search induced by the otherwise uninformative placeholder was sufficient to instantiate neural suppression. This proactive, or predictive, nature of neural suppression guided by implicit prior knowledge supports the growing evidence for predictive and anticipatory processes in visual attention (*Beffara et al., 2023*; *Duncan et al., 2023*; *Serences et al., 2004*; *Zelinsky and Bisley, 2015*). In other words, the brain appears to prepare for potential distractions based on learned probabilities, reflecting a key aspect of predictive processing models. More fundamentally, our results support a perspective of the visual system as relying on prediction to inform perception (*Clark, 2013*; *de Lange et al., 2018*; *Friston, 2005*), echoing previous reports of selective pre-stimulus anticipatory activation in visual cortex (*Kok et al., 2014*).

## Differentiation of implicit statistical learning from volitional attentional strategies

It is well established that attention enhances BOLD responses in visual cortex (*Maunsell, 2015*; *Reynolds and Chelazzi, 2004*; *Williford and Maunsell, 2006*). If participants learned the underlying distractor contingencies, they could deploy an explicit strategy by directing their attention away from the HPDL, e.g., by focusing attention on the diagonally opposite neutral location. This account provides an alternative explanation for the observed EVC modulations. However, while credible, the current findings are not consistent with such an interpretation. First, there was no behavioral facilitation for target stimuli presented at the far neutral location, contrary to what one might expect if participants employed an explicit strategy. However, given the partial dissociation between neural suppression in EVC and behavioral facilitation, additional neural data analyses are required to rule out volitional attention strategies. Thus, we performed a control analysis that excluded all participants that indicated the correct HPDL location in the questionnaire, thereby possibly expressing explicit awareness of the contingencies. This control analysis yielded qualitatively identical results to the full sample, showing significant distractor suppression in EVC. Therefore, it is unlikely that explicit attentional strategies, and the enhancement of locations far from the HPDL, drive the results observed here. Instead, the current findings are consistent with an account emphasizing the automatic deployment of spatial priors (*He et al., 2022*) based on implicitly learned statistical regularities. This account echoes the crucial role of implicit statistical learning (*Kim et al., 2009*; *Turk-Browne et al., 2009*; *Turk-Browne et al., 2010*), adjusting the weights within the proposed spatial priority landscape (*Theeuwes et al., 2022*).

## Differentiation from repetition suppression, priming, and sustained suppression

Repetition suppression (*Henson, 2016*; *Summerfield et al., 2008*; *Todorovic and de Lange, 2012*) and stimulus adaptation (*Adam and Serences, 2021*) must be differentiated from the effects of statistical learning investigated here. Commonly, repetition suppression refers to the attenuation of neural responses, e.g., due to neural adaptation, to repeated presentations of the same or similar stimuli. Crucially, in the present study, stimulus features (orientation and color) were randomized across trials and not related to the key location manipulation. Thus, simple adaptation cannot account for the observed results. However, one could argue that even though visual features do not repeat more frequently, distractors occur more often at the HPDL and hence a higher-level form of repetition

suppression may partially account for the observed results. Several factors argue against this interpretation. First, we report suppression of neural populations in EVC, which is mostly tuned to simple stimulus features, such as orientation (*Hubel and Wiesel, 1962*), which do not repeat more often at the HPDL. Second, we observed suppression irrespective of the stimulus type, including target stimuli, which were equiprobable at all locations, and for neutral stimuli, which were in fact less likely to occur at the HPDL. Finally, during omission trials no stimuli were presented at all, but similar location-specific suppression was observed. Combined these factors rule out an account along the lines of repetition suppression.

Second, participants may have suppressed locations that contained the distractor on the previous trial, reflecting a spatial priming effect. This account constitutes a complementary but different perspective than statistical learning, which integrates implicit prior knowledge across many trials. We ruled out that spatial priming explains the present results by contrasting BOLD suppression magnitudes on trials with the distractor at the HPDL and trials where the distractor was not at the HPDL on the previous trial. Results, depicted in *Figure 4—figure supplement 3*, showed that distractor suppression was statistically significant across both trial types, including trials without a distractor at the HPDL on the preceding trial. This indicates that the observed BOLD suppression is unlikely to be driven by priming and is instead more consistent with statistical learning. Moreover, results did not yield a statistically significant difference between trial types based on the distractor location in the preceding trial. However, these results should not be taken to suggest that spatial priming cannot contribute to distractor suppression; for details, see *Figure 4—figure supplement 3*.

Third, participants might have suppressed the HPDL consistently throughout the experiment. This sustained suppression account differs from the proactive suppression proposed here. While this alternative is plausible, we believe that it is less likely to account for the present results, given the analysis conducted. Specifically, we computed voxelwise parameter estimates and contrasted the obtained betas between locations. Under a sustained suppression account, the HPDL would show suppression even during the implicit baseline period, which would obscure the observed BOLD suppression at and near the HPDL.

## Limitations

Do our modulations in EVC reflect a consequence or a source of spatial priority maps aiding distractor suppression? Our data do not differentiate between these two options. It is conceivable that our results reflect a consequence of downstream instantiations of priority maps, in higher visual areas or beyond, feeding back into EVC; e.g., possibly from parietal cortex (*Bisley and Goldberg, 2010*; *Zelinsky and Bisley, 2015*). Alternatively, our data is also consistent with the notion that EVC itself stores the relevant spatial priority maps, e.g., by short-term synaptic plasticity. However, given the high spatial acuity of EVC representations, it would be surprising if priority maps represented in EVC have the broad spatial characteristics as observed here. Thus, while future research is required to address this question in more detail, our data provide tentative evidence that modulations in EVC, due to learned distractor suppression, are potentially a consequence of downstream instantiated priority maps. Priority signals from these maps then feed back to EVC and hence may lack spatial acuity usually associated with EVC representations.

Are representations of priority signals uniform across EVC? A priori we did not have any hypotheses regarding distinct neural suppression profiles across different early visual areas, hence our primary analyses focused stimulus responses neural populations in EVC, irrespective of subdivision. However, an exploratory analysis suggests that distractor suppression may show different patterns in V1 compared to V2 (Appendix 1). In brief, results in V2 mirrored those reported for the combined EVC ROI (*Figure 4*). In contrast, results in V1 appeared to be only partially modulated by distractor contingencies, and if so, the modulation was less robust and not as spatially broad as in V2. This suggests the possibility of different effects of distractor predictability across subdivisions of early visual areas. However, these results should be interpreted with caution. First, our design did not optimize the delineation of early visual areas (e.g. no functional retinotopy), limiting the accuracy of V1 and V2 segmentation. Additionally, analyses were conducted in volumetric space, which further reduces spatial precision. Future studies could improve this by including retinotopy runs to accurately delineate V1, V2, and V3, and by performing analyses in surface space. Higher-resolution functional

and anatomical MRI sequences would also help elucidate how distractor suppression is implemented across EVC with greater precision.

Given the utilized rapid event-related design, one may question whether our results during omission trials partially reflect lingering activity from preceding search trials, thereby questioning our interpretation that search anticipation alone triggers the deployment of spatial priority maps in EVC. While it is impossible to rule out this possibility completely, we believe that it is unlikely to account for the observed results. First, as omission trials were randomly intermixed with search trials, separated by variable ITIs, and modeled as separate regressors, it is possible to well estimate unique contributions of the omission trials to the BOLD signal. Indeed, the parameter estimates for omission trials were close to zero (i.e. implicit baseline, no visual stimulation), supporting the validity of our inference. Additionally, trials started with the presentation of the placeholder display, which would, if anything, drive neural activity during the omission time window.

Furthermore, we may question whether neural activation during omission trials truly reflects the consequences of the anticipatory deployment of spatial priority maps or a prediction error in response to the surprising omission of the search display. The prediction of a stimulus appearing is equally violated for all locations; hence, we believe that a prediction error should be uniform across the search display. In fact, we might expect larger prediction errors at the HPDL as more reliable priors can be formed about possible inputs at this location (i.e. distractors are more likely to appear). Thus, it seems unlikely that surprise related to the omission of the search display itself can account for the present results.

Finally, due to the limited temporal resolution of BOLD data, the present data do not elucidate whether the present suppression is preceded by a brief attentional enhancement of the HPDL, as implied by some prior work (*Huang et al., 2024*). On this account the HPDL would see transient enhancement, followed by sustained suppression, akin to a 'search and destroy' mechanism. Critically, we believe that this variation would nonetheless constitute proactive distractor suppression as the suppression would still arise before search onset. Using temporally and spatially resolved methods to explore potential transient enhancements preceding suppression is a promising avenue for future research charting the neural mechanisms underlying distractor suppression.

## Conclusion

Our findings elucidate that distractor suppression is a multifaceted process involving both broad, proactive neural mechanisms within the early visual system and more focused downstream modulations. The broad suppression in EVC may serve as an initial filter or bias, which is then fine-tuned by higher (cognitive) processes downstream. Thereby our results indicate that the brain's predictive mechanisms, informed by statistical learning, do not always manifest with spatial precision but can exhibit a broader scope. This phenomenon could represent a neural efficiency trade-off, where the brain balances the cost of precise suppression against the resulting benefits for attentional control. Alternatively, feedback mechanisms, which may signal the hypothesized spatial priority maps from higher-level to lower-level visual areas, could lack the spatial acuity to effectuate more precise location-specific suppression in EVC. Future work is required to assess these possibilities. Our pivotal finding, that suppression is detectable even during omission trials, strongly supports the proactive nature of the deployment of spatial priority maps in guiding visual search, as also suggested by recent behavioral studies (*Huang et al., 2021*; *Huang et al., 2022*; *Kong et al., 2020*). In sum, the broad, proactive nature of neural suppression, alongside its stimulus-unspecific characteristics, suggests a complex interplay between low-level visual processing and higher-order cognitive functions in attentional control to shape spatial priority maps underlying visual search. Finally, our results underscore the significant role of statistical learning and predictive processing in modulating neural responses within the visual cortex, thereby facilitating efficient visual processing amidst the plethora of distracting stimuli impinging upon our senses.

## Materials and methods
### Participants and data exclusion
We acquired MRI data from 32 human volunteers. Data from one participant was lost due to technical problems. Data from three participants were excluded from analysis due to excessive motion

during MRI scanning, defined as an average framewise displacement exceeding 2 SD above the group median. Data from the remaining participants were analyzed, resulting in a final sample size of n=28, satisfying our a priori defined minimum required sample size of n ≥ 24 (based on an a priori power calculation to obtain a power of 80% to detect an effect size Cohen's $d$=0.5 at a default alpha = 0.05). All participants included in the data analysis showed reliable task performance, evident as each subject's response accuracy >82% (group mean 93%) and mean RT for each participant <1050 ms (all trials; group mean 858 ms).

The study followed institutional guidelines and was approved by the local ethics committee (Vaste Commissie Wetenschap en Ethiek of the Vrije Universiteit Amsterdam: VCWE-2021-208R1). Written informed consent and MRI compatibility screening was obtained before study participation. Participants were compensated 15€/hr and were fully debriefed after completion of the experiment.

### Missing data

Data from one main task run (of five runs per participant) was missing for two participants due to technical problems with the MR system. Data from the remaining runs of these participants were included in all analyses.

### Stimuli and experimental paradigm

#### Search task

Participants performed a variant of the additional singleton task (*Theeuwes, 1991*; *Theeuwes, 1992*). Trial onset was signaled by a placeholder display, shown for 800 ms, consisting of eight light gray circles. Each circle had a radius of 1°, arranged in a large circle (5° radius) around the fixation dot. Next, following a brief interstimulus interval of 400 ms, a search display was shown. The search display, presented for 1500 ms, consisted of eight bar stimuli at the same positions as the placeholder circles. Each bar stimulus had a length of 2 DVA and half of the bars contained a small black dot at the center (0.5 DVA). Bar stimuli could either be oriented horizontal or vertical, with all bars having the same orientation, except for one bar stimulus. The one bar with a different orientation constituted the target stimulus. Participants were instructed to report, as quickly and accurately as possible, whether the target stimulus contained a dot at the center or not. Responses were given by button press using the right index and middle finger (dot present/absent). Button mapping was counterbalanced across participants. Target orientation and dot presence/absence did not correlate and could not be predicted, because both target location and dot presence/absence were pseudorandomized. The constraint was that targets appeared equally often at each location of interest in a run (14 times per location; for more details, see *Statistical regularities*).

Additionally, on most trials (42 of 56 search trials per run), a distractor stimulus was shown. The distractor was one of the remaining seven nontarget bar stimuli, but had a different color compared to the other bars. Stimuli could be green or orange, and the distractor always had the other color, thereby making it highly salient. Color was randomized across trials and thus could not be predicted. Trials ended with a variable intertrial interval (ITI) of ~4000 ms (range 3000–8000 ms) during behavioral training and ~5000 ms (range 3000–12,000 ms) during MRI scanning.

Throughout the experiment a white central fixation dot was presented with a diameter of 0.2 DVA. Surrounding the central fixation dot was an outer fixation dot with a diameter of 0.6 DVA and a cross of 0.6 DVA overlayed at its center (see *Thaler et al., 2013*). To provide spatial landmarks on the dark-gray background, a mid-gray octagonal frame (8 DVA radius) was presented around the relevant search display area throughout the experiment.

#### Omission trials

On 25% of trials (19 trials per run) instead of a search display, only a gray screen, identical to the interstimulus interval, was shown for 1500 ms. These omission trials thus contained no visual stimulation, except for the placeholder display and required no response. Omission trials served to probe potential proactive (predictive) deployments of the spatial priority maps hypothesized to underlie distractor suppression. Participants may proactively deploy the spatial priority maps, because on most trials (75%) the appearance of a placeholder display cued the subsequent onset of a search display after 400 ms. Thus, the placeholder can also be thought of as a retrieval or readiness cue. Importantly, placeholder displays were not informative about whether a search display or omission followed on

this specific trial. In the case of a search display trial, placeholders were also not informative about target or distractor location, thus providing no additional usable information, except for signaling likely search onset.

### Statistical regularities

Target location, orientation, color, and dot presence/absence were pseudorandomized and hence not predicable. However, while eight bar stimuli were presented on each trial, target and distractor stimuli could only occur at four specific locations. Thus, four locations were of no interest and always contained neutral stimuli, serving as filler to make the visual search more challenging. The locations of interest (i.e. those that could contain target and distractor stimuli) were fixed for each participant and always had one neighboring location of no interest on each side; e.g., if (going clockwise) location 1 was a location of interest, then location 2 was not a location of interest, but location 3 was, and so on. Across participants locations were pseudorandomized, approximating a balanced number of HPDL locations in the full sample. Note, due to dropout and rejection of participants the final HPDL locations were not fully counterbalanced. Because only four locations could contain target or distractor stimuli and target location was not predictable, targets appeared with 25% chance at each of the four locations of interest.

One of the four locations of interest was designated the HPDL, which contained distractor stimuli (unique color) four times more often than any of the remaining three locations of interest. In other words, if a distractor was present on a given trial (42 trials per run), the distractor appeared 57% (24 trials per run) at the HPDL and at one of the other three locations with equal probability (i.e. 14% or 6 trials per run per location). We refer to these remaining three locations as neutral locations. These neutral locations of interest can be subdivided into two types, the neutral location diagonally across from the HPDL and thus furthest away from the HPDL (NL-far), and the two neutral locations closer to the HPDL (i.e. 90 degrees around the circle; NL-near). The only difference between these two types of neural locations was the distance from the HPDL, in all other aspects these neutral locations were identical to one another. There were no additional statistical regularities governing the characteristics of the stimuli. Only distractor location was predictable, as noted in the paragraph above.

### Behavioral practice

Participants performed three runs of the search task outside of the MRI scanner. The initial two runs served as practice runs, ensuring that participants were familiarized with the task, as the experiment can initially be challenging. These two initial runs did not contain any statistical regularities governing target or distractor locations, thereby precluding any learning during these initial practice runs. Only the final, third run in the behavioral lab contained the statistical regularities (HPDL) as described above. Except for a shorter ITI and slightly fewer omission trials (9 instead of 19 trials), the third behavioral practice run was identical to the MRI runs.

Additionally, the behavioral runs served to train participants to maintain fixation on the central fixation dot throughout the experiment. Specifically, using an eyetracker (Eyelink-1000), we provided gaze contingent feedback to participants, indicating when fixation was broken, defined as a fixation >1.5 DVA from the central fixation dot. Feedback on breaking fixation was provided by flashing a red octagon (stop sign; 3.5 DVA in size) at the center of the screen. The importance of maintaining fixation throughout the run was emphasized to participants. While no gaze contingent feedback was provided during subsequent MRI scanning, participants were informed that experimenters monitored their fixation behavior and were reminded between runs to maintain fixation.

### Procedure

Following the three behavioral practice runs in the behavioral lab, participants enter the MRI scanner. Here, they performed five runs of the additional singleton task with distractor contingencies as outlined before, with each run consisting of 72 trials each (~9 min per run). Following these main task runs, participants answered a brief questionnaire assessing their explicit awareness of the statistical regularities determining distractor appearances. Next, participants performed two location localizer runs, outlined below. Finally, anatomical scans (T1) were acquired, and participants were debriefed.

## Location localizer

To select neural populations (voxels) with their receptive fields corresponding to each of the locations of interest (HPDL, NL-far, twice NL-near), we performed a location localizer. This localizer consisted of a block design, displaying a checkerboard cross stimulus at one of the locations of interest, with the stimulus sizes matching the same dimensions as the bar stimuli. Black and white inverted at a rate of 4 Hz. The checkboard cross stimulus stayed at each location of interest for 12 s, before randomly switching to one of the other three locations of interest. Each location was sampled 8 times per run. Additionally, two null events (only fixation) of 12 s were added per run. Participants performed two runs of this localizer and were tasked to press the index finger button whenever the central fixation dot turned red. Timing of the color change of the fixation dot could not be predicted and occurred randomly between 3 and 9 s during a block. Thus, during this localizer the stimuli and task were different compared to the search task, only one checkerboard stimulus was presented at a time, and there were no statistical regulations governing where the stimulus was shown next (except for avoiding direct repetitions of the same location).

## Questionnaire

Following completion of the search task participants completed a brief on-screen questionnaire. The key question probed whether participants noticed the statistical regularities governing the occurrence of the distractor stimuli. Participants were presented with a display similar to a placeholder display (i.e. white circles corresponding to the eight stimulus locations) and asked to indicate the HPDL. Precise instructions were: 'Here you see the layout of the experiment. Select the location that was most likely to show a distractor (unique color). Give your answer by moving the red diamond to the one location that was most likely to have a distractor'. Participants controlled the location of a red diamond, using their index and middle finger buttons, and then confirmed their selection of the HPDL using the ring finger button. They could go back to the question should they have made an accidental response.

## Stimulus presentation

Stimuli were presented on an MRI-compatible 32" BOLD screen from Cambridge Research Systems, visible via an adjustable mirror mounted on the headcoil. In the behavioral lab stimuli were presented on a Display++ screen from Cambridge Research Systems.

## MRI data acquisition

Data was acquired on a 3 Tesla Philips Ingenia CX MRI Scanner using a 32-channel headcoil. Functional images were acquired using a T2*-weighted EPI sequence with the following parameters: TR/TE = 1600/30 ms, 56 slices, voxel size 2 mm isotropic, FOV = $224 \times 224 \times 123$ mm$^3$, flip angle = 70°, AP fold-over direction, SENSE P reduction (AP)=1.5, multiband factor = 4, bandwidth = 1911.7 Hz. Anatomical images were acquired using a T1-weighted 3D fast field echo sequence with the following parameters: TR/TE = 7.0/3.2 ms, voxel size 1 mm isotropic, FOV = $256 \times 256 \times 176$ mm$^3$, flip angle = 8°, sagittal slice orientation, using CS-SENSE with a reduction factor = 6.

## Statistical analysis

### Behavioral data analysis

Data was analyzed in terms of RT and response accuracy. Trials with responses faster than 100 ms or slower than 1500 ms (i.e. after search display offset) were rejected from analysis. For RT analyses only correct response trials were analyzed. Two separate behavioral analyses were performed. The first analysis was distractor contingent, splitting up trials depending on where the distractor occurred on a given trial (levels: distractor absent, distractor at the HPDL, NL-near, or NL-far location). The second analysis was target contingent, splitting trials by where the target appeared on the trial (levels: target at HPDL, NL-near, or NL-far). Average RTs and response accuracies were calculated for each participant and subjected to separate one-way repeated measures ANOVAs with four levels (distractor contingent) or three levels (target contingent). Following significant results in the repeated measures ANOVAs, post hoc pairwise comparisons were conducted using t-tests or Wilcoxon signed-rank tests as appropriate. p-Values from these t-tests were corrected using the Holm-Bonferroni method. Effect size estimates were calculated for t-tests as Cohen's d (*Lakens, 2013*), Wilcoxon signed-rank tests as

matched-pairs rank-biserial correlation ($r$), and partial eta-squared ($\eta_p^2$) for repeated measures ANOVAs. Within-subject standard errors of the mean, as depicted in *Figures 4 and 2* and *Figure 5—figure supplement 1* were calculated using the within-subject normalization procedure by *Cousineau, 2005*, with *Morey, 2008*, bias correction. Bayesian analyses were performed to evaluate evidence for the absence of an effect of target location (*Figure 5—figure supplement 1*). We used JASP 0.18.2 (*JASP Team, 2023*) with default settings for Bayesian t-tests with a Cauchy prior width of 0.707. Resulting Bayes factors were interpreted qualitatively based on *Lee and Wagenmakers, 2014*.

## fMRI data preprocessing

MRI data was preprocessed using FSL 6.0.6.5 (FMRIB Software Library; Oxford, UK; http://www.fmrib.ox.ac.uk/fsl; RRID:SCR_002823; *Smith et al., 2004*). The first five volumes of each run were discarded to allow for signal stabilization. Data processing included brain extraction using BET, motion correction using MCFLIRT, temporal high pass filtering at a 128 s cutoff, and spatial smoothing at 5 mm fwhm. fMRI images were registered to the anatomical (T1) image using Boundary-Based Registration (BBR) as implemented in FSL FLIRT and subsequently normalized to the MNI152 T1 2 mm template with linear registration (12 DF).

## fMRI data analysis

fMRI data was analyzed using FSL FEAT, fitting each subject's run data using voxelwise general linear models (GLMs) in an event-related approach. To obtain contrast parameter estimates for each stimulus type at each location, we included regressors for all target and distractor location combinations (e.g. target location 1 and distractor absent; target location 1 and distractor location 2; target location 1 and distractor location 3; and so on), as well as one regressor for omission trials in the first level GLMs. Nuisance regressors of no interest were added to the GLM, including first-order temporal derivatives for all modeled regressors and 24 motion regressors, comprised of the standard+extended set of motion parameters of FSL FEAT (i.e. six standard motion parameters, their temporal derivatives, the squared motion parameters, and the squared derivatives of the motion parameters). Across runs data was combined using FSL's fixed effects analysis.

We defined all stimulus types and location combinations as contrasts, i.e., targets, distractors, and neutral stimuli at each of the four locations, thus resulting in 12 contrasts of interest. From these contrast parameter estimates we extracted the conditions of interest, which were analyzed using an ROI-based approach (see below for details on the ROI definition). Specifically, for search trials we analyzed the extracted parameter estimates within the ROIs using a 3 by 3 repeated measures ANOVA with stimulus type (target, distractor, neutral stimuli) and location (HPDL, NL-near, NL-far) as factors. Note that NL-near is represented by two location contrasts, thus parameter estimates for this contrast were averaged. For omission trials we analyzed the parameter estimates within the ROIs using a one-way repeated measures ANOVA with location (HPDL, NL-near, NL-far) as factor. As for the behavioral analysis, statistically significant results in the repeated measures ANOVAs were followed up using post hoc pairwise comparisons. For the search trial analysis, we also performed planned paired t-tests contrasting the response to the three locations within each stimulus type (i.e. targets at HPDL vs targets at NL-far, and so on). Again, we report effect sizes for all contrasts (see *Behavioral data analysis* for details).

## ROI definition

ROIs were neural populations in EVC with receptive fields corresponding to the four locations of interest (HPDL, NL-far, and the two NL-near locations). To derive these ROIs, we first defined an anatomically derived EVC mask, comprising V1, V2, and V3, inspired by prior studies showing consistent effects of distractor suppression (*Adam and Serences, 2021*). To this end we extracted V1 and V2 (*Amunts et al., 2000*), as well as V3 (*Rottschy et al., 2007*) labels, for both left and right hemispheres, from the Jülich-Brain Cytoarchitectonic Atlas (*Amunts et al., 2020*) as distributed with FSL. Next, we combined V1, V2, and V3 into one combined, bilateral EVC mask. This mask was then subdivided and constrained for each participant separately using individual location localizer run data. Specifically, we selected the voxels most responsive to each visually stimulated location compared to the other three locations. For example, the EVC location 1 mask were the 20 voxels most responsive in the contrast: location 1>mean(location 2+3+4). Using the location localizer allowed us to independently establish

the ROI masks from the main task data and refine the location masks for each participant individually. We validated the reliability of the location selective masks by cross-validating the mask selectivity using only one run of the localizer and testing on the other localizer run. Results confirmed reliable location ROI definition (*Figure 4—figure supplement 4*). Additionally, zero voxels overlapped across the location-specific masks up until a mask size of 400 voxels, thus further establishing location mask selectivity.

## Acknowledgements

We thank Carlota Sabate, Eleonora Assarioti, and Mayca Thijssen for help with data acquisition, as well as Wietske van der Zwaag, Diederick Stoffers, and Tomas Knapen for assistance with technical, MR sequence, and administrative questions. This work was supported by the Nederlandse Organisatie voor Wetenschappelijk Onderzoek (NWO) SSH Open Competition Behaviour and Education 2021 grant (Reference number: 406.21.GO.034, '*Learning to direct attention in space and time*') awarded to JT.

## Additional information

### Funding

| Funder | Grant reference number | Author |
|---|---|---|
| Nederlandse Organisatie voor Wetenschappelijk Onderzoek | 406.21.GO.034 | Jan Theeuwes |

The funders had no role in study design, data collection and interpretation, or the decision to submit the work for publication.

### Author contributions

David Richter, Conceptualization, Resources, Data curation, Software, Formal analysis, Validation, Investigation, Visualization, Methodology, Writing – original draft, Project administration, Writing – review and editing; Dirk van Moorselaar, Conceptualization, Methodology, Writing – review and editing; Jan Theeuwes, Conceptualization, Supervision, Funding acquisition, Methodology, Writing – review and editing

### Author ORCIDs

David Richter ⓘ https://orcid.org/0000-0002-3404-8374
Dirk van Moorselaar ⓘ https://orcid.org/0000-0002-0491-1317

### Ethics

The study followed institutional guidelines and was approved by the local ethics committee (Vaste Commissie Wetenschap en Ethiek of the Vrije Universiteit Amsterdam: VCWE-2021-208R1). Written informed consent and MRI compatibility screening was obtained before study participation.

Reviewer #1 (Public review): https://doi.org/10.7554/eLife.101733.3.sa1
Reviewer #2 (Public review): https://doi.org/10.7554/eLife.101733.3.sa2
Author response https://doi.org/10.7554/eLife.101733.3.sa3

## Additional files

### Supplementary files

Supplementary file 1. Results of planned pairwise tests contrasting fMRI BOLD responses during search trials. Contrasted are the three stimulus locations (high-probability distractor location [HPDL], neutral location nearby the HPDL [NL-near], neutral location furthest away from the HPDL [NL-far]) for each stimulus type (distractor, target, neutral stimulus) separately. Reported are paired t-tests or Wilcoxon signed-rank tests results as appropriate with associated effect sizes (Cohen's d for t-tests

and matched rank biserial correlation for Wilcoxon signed-rank tests). p-Values are uncorrected. Bayes factors denote the $BF_{10}$ from Bayesian paired t-tests.

MDAR checklist

## Data availability

Analysis and experiment code, along with processed data necessary to replicate the results, are available here: https://doi.org/10.17605/OSF.IO/G4RXV. Additionally, figure source data contains the numerical data used to generate the figures. Due to institutional and EU privacy regulations, raw MRI data are not shared publicly, as standard anonymization may not ensure long-term de-identification. However, access can be granted on a case-by-case basis. Interested researchers may contact the corresponding author to request access.

The following dataset was generated:

| Author(s) | Year | Dataset title | Dataset URL | Database and Identifier |
| --- | --- | --- | --- | --- |
| Richter D, van Moorselaar D, Theeuwes J | 2024 | Data from: Proactive distractor suppression in early visual cortex | https://doi.org/10.17605/OSF.IO/G4RXV | Open Science Framework, 10.17605/OSF.IO/G4RXV |

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

## Appendix 1

In an exploratory analysis we investigated whether subdivisions of EVC exhibit different representations of priority signals. In brief, we used FreeSurfer to reconstruct brain surfaces (recon-all) from each subject's anatomical scan. From these reconstructions we derived V1_exvivo and V2_exvivo labels, which were transformed into volume space using 'mri_label2vol' and merged into a bilateral mask for each ROI. We then selected the voxels within each ROI that were most responsive to the four stimulus locations, based on independent localizer data. This voxel selection followed the procedure outlined in the Materials and methods: ROI definition. To accommodate the subdivision into two ROIs (V1 and V2) compared to the single EVC ROI in the main analysis, we halved the number of voxels selected per location. Finally, we applied the same ROI analysis to investigate distractor suppression during search and omission trials, following the procedure described in Materials and methods: Statistical analysis.

Results of this more fine-grained ROI analyses are depicted in *Appendix 1—figure 1*. First, the results from V2 qualitatively mirrored our primary ROI analysis. BOLD responses in V2 differed significantly between stimulus types (main effect of stimulus type: $F_{(2,54)}$ = 31.11, p<0.001, $\eta_p^2$ = 0.54). Targets elicited larger BOLD responses compared to distractors ($t_{(27)}$ = 3.05, $p_{holm}$ = 0.004, $d$=0.06) and neutral stimuli ($t_{(27)}$ = 7.82, $p_{holm}$<0.001, $d$=0.14). Distractors also evoked larger responses than neutral stimuli ($t_{(27)}$ = 4.78, $p_{holm}$<0.001, $d$=0.09). These results likely reflect top-down modulation due to target relevance and bottom-up effects of distractor salience. Consistent with the primary ROI analysis, the manipulation of distractor predictability showed a distinct pattern of location-specific BOLD suppression in V2 (main effect of location: $F_{(1.1,52.8)}$=5.01, p=0.030, $\eta_p^2$ = 0.16). Neural populations with receptive fields at the HPDL showed significantly reduced BOLD responses compared to the diagonally opposite neutral location (NL-far; post hoc test HPDL vs NL-far: $t_{(27)}$ = 2.69, $p_{holm}$ = 0.022, $d$=0.62). Again, this suppression was not confined to the HPDL but also extended to close by neutral locations (NL-near vs NL-far: $t_{(27)}$ = 2.79, $p_{holm}$ = 0.022, $d$=0.65). BOLD responses did not differ between HPDL and NL-near locations (HPDL vs NL-near: $t_{(27)}$ = 0.11, $p_{holm}$ = 0.915, $d$=0.03; $BF_{10}$=0.13). As in the EVC ROI analysis, this suppression pattern was consistent across distractor, target, and neutral stimuli presented at the HPDL and NL-near locations compared to NL-far. In sum, neural responses in V2 were significantly modulated by the distractor contingencies, evident as reduced BOLD responses in neural populations with receptive fields at the HPDL and neutral locations near the location of the frequent distractor (NL-near), relative to the neutral location diagonally across the HPDL (NL-far).

In V1, BOLD responses also differed significantly between stimulus types (main effect of stimulus type: $F_{(1.3,35.6)}$=6.69, p=0.009, $\eta_p^2$ = 0.20). Targets elicited larger BOLD responses compared to neutral stimuli ($t_{(27)}$ = 3.52, $p_{holm}$ = 0.003, $d$=0.12) and distractors evoked larger responses than neutral stimuli ($t_{(27)}$ = 2.62, $p_{holm}$ = 0.023, $d$=0.09). However, no difference between targets and distractors was observed ($t_{(27)}$ = 0.90, $p_{holm}$ = 0.375, $d$=0.03; $BF_{10}$=0.17), suggesting reduced sensitivity to task-related effects in V1. Indeed, analyzing the effect of distractor predictability for BOLD responses in V1 showed a different result than in V2 and the combined EVC ROI. There was no significant main effect of location ($F_{(2,54)}$ = 2.20, p=0.120, $\eta_p^2$ = 0.08; $BF_{10}$=0.77). BOLD responses at NL-near and NL-far were similar ($BF_{10}$=0.171), with the only reliable difference found between target stimuli at the HPDL and NL-far locations (W=94, $p_{holm}$ = 0.012, $r$=0.54).

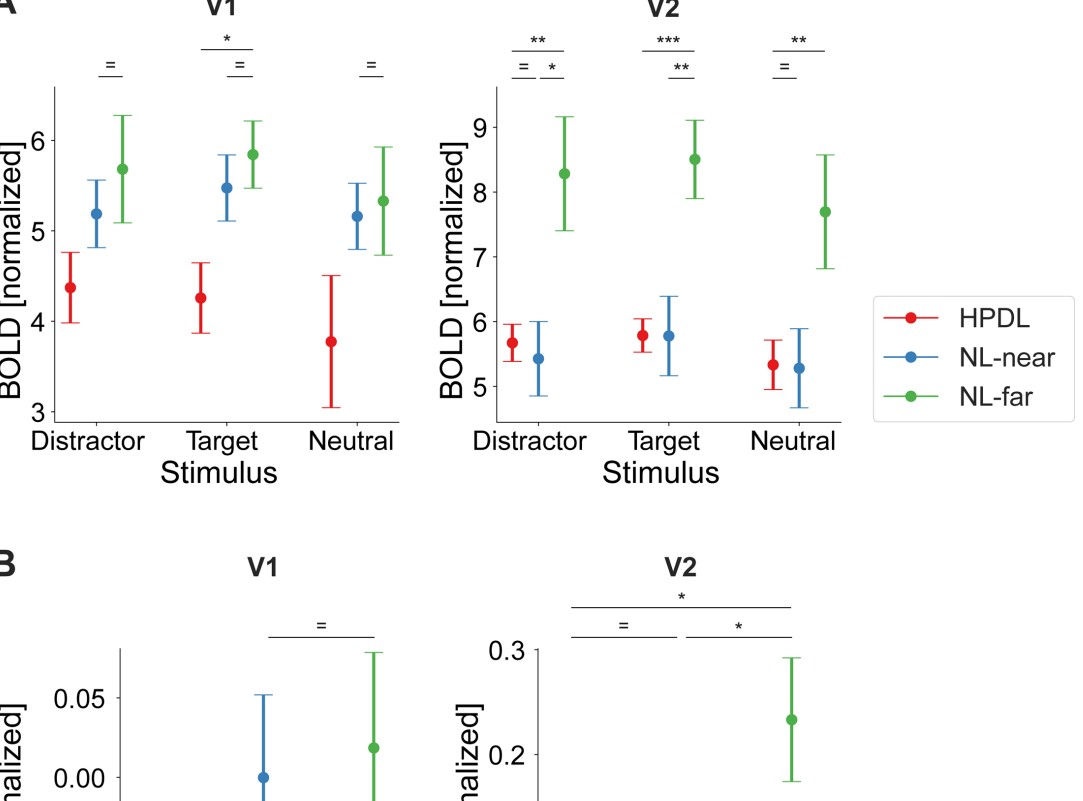

**Appendix 1—figure 1.** Distractor suppression in primary and secondary visual cortex. (**A**) fMRI BOLD responses (ordinate) during search trials in V1 (left) and V2 (right), split into stimulus types (abscissa). Color denotes locations based on distractor contingencies with red = high-probability distractor location (HPDL), blue = neutral locations near the HPDL (NL-near), green = neutral locations far from the HPDL (NL-far; diagonally opposite from the HPDL). (**B**) Corresponding results for omission trials in V1 (left) and V2 (right). Neural populations with receptive fields at the HPDL and NL-near locations were suppressed compared to those with receptive fields at the NL-far location in V2, but not V1. Error bars denote within-subject SEM. Asterisks indicate statistically significant pairwise comparisons within stimulus types: *p<0.05, **p<0.01, ***p<0.001, = $BF_{10}<1/3$.

The online version of this article includes the following source data for appendix 1—figure 1:

**Appendix 1—figure 1—source data 1.** Distractor suppression in primary visual cortex – search trials.

**Appendix 1—figure 1—source data 2.** Distractor suppression in secondary visual cortex – search trials.

**Appendix 1—figure 1—source data 3.** Distractor suppression in primary visual cortex – omission trials.

**Appendix 1—figure 1—source data 4.** Distractor suppression in secondary visual cortex – omission trials.

