## [Editor Report · eLife Assessment]

This **important** and well-written study uses functional neuroimaging in human observers to provide **compelling** evidence that activity in the early visual cortex is suppressed at locations that are frequently occupied by a task-irrelevant but salient item. This suppression appears to be general to any kind of stimulus and also occurs in advance of any item actually appearing. The work will be of great interest to psychologists and neuroscientists examining attention, perception, learning and prediction.

---

## [Referee Report · Reviewer #1 (Public review)]

Summary:

The authors investigated if/how distractor suppression derived from statistical learning may be implemented in early visual cortex. While in a scanner, participants conducted a standard additional singleton task in which one location more frequently contained a salient distractor. The results showed that activity in EVC was suppressed for the location of the salient distractor as well as for neighbouring neutral locations. This suppression was not stimulus specific - meaning it occurred equally for distractors, targets and neutral items - and it was even present in trials in which the search display was omitted. Generally, the paper was clear, the experiment was well-designed, and the data are interesting.

The authors addressed all of my concerns and the revised manuscript will make a beautiful addition to the literature.

---

## [Referee Report · Reviewer #2 (Public review)]

The authors of this work set out to test ideas about how observers learn to ignore irrelevant visual information. Specifically, they used fMRI to scan participants who performed a visual search task. The task was designed in such a way that highly salient but irrelevant search items were more likely to appear at a given spatial location. With a region-of-interest approach, the authors found that activity in visual cortex that selectively responds to that location was generally suppressed, in response to all stimuli (search targets, salient distractors, or neutral items), as well as in the absence of an anticipated stimulus.

Strengths of the study include: A well-written and well-argued manuscript; clever application of a region of interest approach to fMRI design, which allows articulating clear tests of different hypotheses; careful application of follow-up analyses to rule out alternative, strategy-based accounts of the findings; tests of the robustness of the findings to detailed analysis parameters such as ROI size; and exclusion of the role of regional baseline differences in BOLD responses. The main findings are enhanced by supplementary analyses that distinguish between the responses of early visual areas.

The study provides an advance over previous studies, which identified enhancement or suppression in visual cortex as a function of search target/distractor predictability, but in less spatially-specific way. It also speaks to open questions about whether such suppression/enhancement is observed only in response to the arrival of visual information, or instead is preparatory, favouring the latter view. These questions have been at the heart of theoretical debates in this literature on how distractor suppression unfolds in the context of visual search.

---

## [Author Response]

The following is the authors’ response to the original reviews.

**eLife Assessment**
This well-written report uses functional neuroimaging in human observers to provide convincing evidence that activity in the early visual cortex is suppressed at locations that are frequently occupied by a task-irrelevant but salient item. This suppression appears to be general to any kind of stimulus, and also occurs in advance of any item actually appearing. The work in its present form will be valuable to those examining attention, perception, learning and prediction, but with a few additional analyses could more informatively rule out potential alternative hypotheses. Further discussion of the mechanistic implications could clarify further the broad extent of its significance.

We thank the editor and the reviewers for the positive evaluation of our manuscript and the thoughtful comments. Below we provide a detailed point-by-point reply to the reviewers’ comments.

In addition to addressing the reviewers' comments, we have improved the figure legends by explicitly describing the type of error bars depicted in the figures, information which was previously only listed in the Materials and Methods section. Specifically, the statement: “Error bars denote within-subject SEM” was added to several figures, as applicable. We believe that briefly reiterating this information in the figure legends enhances clarity and enables readers to interpret the results more accurately and efficiently. We also updated our code and data sharing statement, as well as opened the repository for the public: “Analysis and experiment code, as well as data required to replicate the results reported in this manuscript are available here: https://doi.org/10.17605/OSF.IO/G4RXV. Raw MRI data is available upon request.”

**Public Reviews**

**Reviewer #1 (Public review):**
Summary:The authors investigated if/how distractor suppression derived from statistical learning may be implemented in early visual cortex. While in a scanner, participants conducted a standard additional singleton task in which one location more frequently contained a salient distractor. The results showed that activity in EVC was suppressed for the location of the salient distractor as well as for neighbouring neutral locations. This suppression was not stimulus specific - meaning it occurred equally for distractors, targets and neutral items - and it was even present in trials in which the search display was omitted. Generally, the paper was clear, the experiment was well-designed, and the data are interesting. Nevertheless, I do have several concerns mostly regarding the interpretation of the results.(1) My biggest concern with the study is regarding the interpretation of some of the results. Specifically, regarding the dynamics of the suppression. I appreciate that there are some limitations with what you might be able to say here given the method but I do feel as if you have committed to a single interpretation where others might still be at play. Below I've listed a few alternatives to consider.

We agree with the reviewer that there are important alternatives to consider. Adequately addressing these alternatives will substantially increase the inferences we can draw from our data. Therefore, we address each alternative interpretation in detail below.

(a) Sustained Suppression. I was wondering if there is anything in your results that would speak for or against the suppression being task specific. That is, is it possible that people are just suppressing the HPDL throughout the entire experiment (i.e., also through ITI, breaks, etc., rather than just before and during the search). Since the suppression does not seem volitional, I wonder if participants might apply a blanket suppression to HPDL un l they learn otherwise. Since your localiser comes a er the task you might be able to see hints of sustained suppression in the HPDL during these trials.

It is indeed possible that participants suppressed the HPDL throughout the entire experiment, instead of proactively instantiating suppression on each trial. While possible, we believe that this account is less likely to explain the present results, given the utilized analysis approach, a voxel-wise GLM fit to the BOLD data per run (see Materials and Methods for details). Specifically, we derived parameter estimates from this GLM per location to estimate the relative suppression. Sustained suppression would modulate BOLD responses throughout the run, i.e. presumably also during the implicit baseline period used to estimate the contrast parameter estimates per location. Hence, sustained suppression should not result in a differential modulation between locations, as the BOLD response at the HPDL during the baseline period would be equally suppressed as during the trial. Inspired by the reviewer’s comment, we now clarify this critical point in the manuscript’s Discussion section:

“Third, participants might have suppressed the HPDL consistently throughout the experiment. This sustained suppression account differs from the proactive suppression proposed here. While this alternative is plausible, we believe that it is less likely to account for the present results, given the analysis conducted. Specifically, we computed voxel-wise parameter estimates and contrasted the obtained betas between locations. Under a sustained suppression account, the HPDL would show suppression even during the implicit baseline period, which would obscure the observed BOLD suppression at and near the HPDL.”

(b) Enhancement followed by suppression. Another alternative that wasn't discussed would be an initial transient enhancement of the HPDL which might be brought on by the placeholders followed by more sustained suppression through the search task. Of course, on the whole this would look like suppression, but this still seems like it would hold different implications compared to simply "proactive suppression". This would be something like search and destroy however could be on the location level before the actual onset of the search display.

R1 correctly points out that BOLD data, given the poor temporal resolution, do not allow for the detection of potential transient enhancements at the HPDL followed by a later and more pronounced suppression (akin to “search and destroy”). We fully agree with this assessment. However, we also argue that a transient enhancement followed by sustained suppression before search display onset constitutes proactive suppression in line with our interpretation, because suppression would still arise proactively (i.e., before search, and hence distractor, onset). Whether transient enhancement precedes suppression cannot be elucidated by our data, but we believe that it constitutes an interesting avenue for future studies using me-resolved and spatially specific recording methods. We now clarify this important implementational variation in the updated manuscript.

“Finally, due to the limited temporal resolution of BOLD data, the present data do not elucidate whether the present suppression is preceded by a brief attentional enhancement of the HPDL, as implied by some prior work (Huang et al., 2024). On this account the HPDL would see transient enhancement, followed by sustained suppression, akin to a ‘search and destroy’ mechanism. Critically, we believe that this variation would nonetheless constitute proactive distractor suppression as the suppression would still arise before search onset. Using temporally and spatially resolved methods to explore potential transient enhancements preceding suppression is a promising avenue for future research charting the neural mechanisms underlying distractor suppression.”

(2) I was also considering whether your effects might be at least partially attributable to priming type effects. This would be on the spatial (not feature) level as it is clear that the distractors are switching colours. Basically, is it possible that on trial n participants see the HPDL with the distractor in it and then on trial n+1 they suppress that location. This would be something distinct from the statistical learning framework and from the repetition suppression discussion you have already included. To test for this, you could look at the trials that follow omission or trials. If there is no suppression or less suppression on these trials it would seem fair to conclude that the suppression is at least in part due to the previous trial.

We agree with the reviewer that it is plausible that participants particularly suppress locations which on previous trials contained a distractor. To address this possibility, we conducted a new analysis and adjusted the manuscript accordingly:

“Second, participants may have suppressed locations that contained the distractor on the previous trial, reflecting a spatial priming effect. This account constitutes a complementary but different perspective than statistical learning, which integrates implicit prior knowledge across many trials. We ruled out that spatial priming explains the present results by contrasting BOLD suppression magnitudes on trials with the distractor at the HPDL and trials where the distractor was not at the HPDL on the previous trial. Results, depicted in Supplementary Figure 4 showed that distractor suppression was statistically significant across both trial types, including trials without a distractor at the HPDL on the preceding trial. This indicates that the observed BOLD suppression is unlikely to be driven by priming and is instead more consistent with statistical learning. Moreover, results did not yield a statistically significant difference between trial types based on the distractor location in the preceding trial. However, these results should not be taken to suggest that spatial priming cannot contribute to distractor suppression; for details see: Supplementary Figure 4.” (p. 13).

We note that this analysis approach slightly differs from the reviewer’s suggestion, which considered omission trials. However, we decided to exclude trials immediately following an omission to ensure that both conditions were matched as closely as possible. In particular, omission trials represent extended rest periods, which could alter participants’ state and especially modulate the visually evoked BOLD responses (e.g., potentially increasing the dynamic range) compared to trials that did not follow omissions. Our analysis approach avoids this difference while still addressing the hypothesis put forward by the reviewer. We now provide the full explanation and results figure of this priming analysis in the figure text of Supplementary Figure 4:

**Reviewer #2 (Public review):**
The authors of this work set out to test ideas about how observers learn to ignore irrelevant visual information. Specifically, they used fMRI to scan participants who performed a visual search task. The task was designed in such a way that highly salient but irrelevant search items were more likely to appear at a given spatial location. With a region-of-interest approach, the authors found that activity in visual cortex that selectively responds to that location was generally suppressed, in response to all stimuli (search targets, salient distractors, or neutral items), as well as in the absence of an anticipated stimulus.Strengths of the study include: A well-written and well-argued manuscript; clever application of a region of interest approach to fMRI design, which allows articulating clear tests of different hypotheses; careful application of follow-up analyses to rule out alternative, strategy-based accounts of the findings; tests of the robustness of the findings to detailed analysis parameters such as ROI size; and exclusion of the role of regional baseline differences in BOLD responses.

We thank the reviewer for the positive evaluation of our manuscript.

The report might be enhanced by analyses (perhaps in a surface space) that distinguish amongst the multiple "early" retinotopic visual areas that are analysed in the aggregate here.

We agree with the reviewer that an exploratory analysis separating early visual cortex (EVC) into its retinotopic areas could be an interesting addition. Our reasoning to combine early visual areas into one mask in the original analyses was two-fold: First, we did not have an a priori reason to expected distinct neural suppression between these early ROIs. Therefore, we did not acquire retinotopy data to reliably separate early visual areas (e.g. V1, V2 and V3), instead opting to increase the number of search task trials. The lack of retinotopy data inherently limits the reliability of the resulting cortical segmentation. However, we now performed an analysis separating early visual cortex into V1 and V2 and report the details as Supplementary Text 1:

“In an exploratory analysis we investigated whether subdivisions of EVC exhibit different representations of priority signals. In brief, we used FreeSurfer to reconstruct brain surfaces (recon-all) from each subject’s anatomical scan. From these reconstructions we derived V1_exvivo and V2_exvivo labels, which were transformed into volume space using ‘mri_label2vol’ and merged into a bilateral mask for each ROI. We then selected the voxels within each ROI that were most responsive to the four stimulus locations, based on independent localizer data. This voxel selection followed the procedure outlined in the Materials and Methods: Region of Interest (ROI) Definition. To accommodate the subdivision into two ROIs (V1 and V2) compared to the single EVC ROI in the main analysis, we halved the number of voxels selected per location. Finally, we applied the same ROI analysis to investigate distractor suppression during search and omission trials, following the procedure described in Materials and Methods: Statistical Analysis.

Results of this more fine-grained ROI analyses are depicted in Supplementary Figure 1. First, the results from V2 qualitatively mirrored our primary ROI analysis. BOLD responses in V2 differed significantly between stimulus types (main effect of stimulus type: F_(2,54)_ = 31.11, p < 0.001, 𝜂 = 0.54). Targets elicited larger BOLD responses compared to distractors (t_(27)_ = 3.05, p_holm_ = 0.004, d = 0.06) and neutral stimuli (t_(27)_ = 7.82, p_holm_ < 0.001, d = 0.14). Distractors also evoked larger responses than neutral stimuli (t_(27)_ = 4.78, p_holm_ < 0.001, d = 0.09). These results likely reflect top-down modulation due to target relevance and bo om-up effects of distractor salience. Consistent with the primary ROI analysis, the manipula on of distractor predictability showed a distinct pattern of location specific BOLD suppression in V2 (main effect of location: F_(1.1,52.8)_ = 5.01, p = 0.030, 𝜂 = 0.16). Neural populations with receptive fields at the HPDL showed significantly reduced BOLD responses compared to the diagonally opposite neutral location (NL-far; post hoc test HPDL vs NL-far: t_(27)_ = 2.69, p_holm_ = 0.022, d = 0.62). Again, this suppression was not confined to the HPDL but also extended to close by neutral locations (NL-near vs NL-far: t_(27)_ = 2.79, p_holm_ = 0.022, d = 0.65). BOLD responses did not differ between HPDL and NL-near locations (HPDL vs NL-near: t_(27)_ = 0.11, p_holm_ = 0.915, d = 0.03; BF_10_ = 0.13). As in the EVC ROI analysis, this suppression pattern was consistent across distractor, target, and neutral stimuli presented at the HPDL and NL-near locations compared to NL-far. In sum, neural responses in V2 were significantly modulated by the distractor contingencies, evident as reduced BOLD responses in neural populations with receptive fields at the HPDL and neutral locations near the location of the frequent distractor (NL-near), relative to the neutral location diagonally across the HPDL (NL-far).

In V1, BOLD responses also differed significantly between stimulus types (main effect of stimulus type: F_(1.3,35.6)_ = 6.69, p = 0.009, 𝜂 = 0.20). Targets elicited larger BOLD responses compared neutral stimuli (t_(27)_ = 3.52, p_holm_ = 0.003, d = 0.12) and distractors evoked larger responses than neutral stimuli (t_(27)_ = 2.62, p_holm_ = 0.023, d = 0.09). However, no difference between targets and distractors was observed (t_(27)_ = 0.90, p_holm_ = 0.375, d = 0.03; BF_10_ = 0.17), suggesting reduced sensitivity to task-related effects in V1. Indeed, analyzing the effect of distractor predictability for BOLD responses in V1 showed a different result than in V2 and the combined EVC ROI. There was no significant main effect of location (F_(2,54)_ = 2.20, p = 0.120, 𝜂 = 0.08; BF_10_ = 0.77). BOLD responses at NL-near and NL-far were similar (BF_10_ = 0.171), with the only reliable difference found between target stimuli at the HPDL and NL-far locations (W = 94, p_holm_ = 0.012, r = 0.54).”

We include the new result figure as Supplementary Figure 5

We now include reference to these results in the manuscript’s Discussion section:

“Are representations of priority signals uniform across EVC? A priori we did not have any hypotheses regarding distinct neural suppression profiles across different early visual areas, hence our primary analyses focused stimulus responses neural populations in EVC, irrespective of subdivision. However, an exploratory analysis suggests that distractor suppression may show different patterns in V1 compared to V2 (Supplementary Figure 5 and Supplementary Text 1). In brief, results in V2 mirrored those reported for the combined EVC ROI (Figure 4). In contrast, results in V1 appeared to be only partially modulated by distractor contingencies, and if so, the modulation was less robust and not as spatially broad as in V2. This suggests the possibility of different effects of distractor predictability across subdivisions of early visual areas. However, these results should be interpreted with caution. First, our design did not optimize the delineation of early visual areas (e.g., no functional retinotopy), limiting the accuracy of V1 and V2 segmentation. Additionally, analyses were conducted in volumetric space, which further reduces spatial precision. Future studies could improve this by including retinotopy runs to accurately delineate V1, V2, and V3, and by performing analyses in surface space. Higher-resolution functional and anatomical MRI sequences would also help elucidate how distractor suppression is implemented across EVC with greater precision.”

Furthermore, the study could benefit from an analysis that tests the correlation over observers between the magnitude of their behavioural effects and their neural responses.

R2 highlights that behavioral facilitation and neural suppression could be correlated across participants. The rationale is that if neural suppression in EVC is related to the facilitation of behavioral responses, we should expect a positive relationship between neural suppression at the HPDL and RTs across participants. In this analysis we focused on the contrast between HPDL and NL-far, as this contrast was statistically significant in both the RT (Figure 2) and the neural suppression analysis (Figure 4). First, we computed for each participant the behavioural benefit of distractor suppression as: RT_facilitation_ = RT_NL-far_ – RT_HPDL_. Thereby RT facilitation reflects the response speeding due to a distractor appearing at the high probability distractor location compared to the far neutral location. Next, we computed neural suppression as: BOLD_suppression_ = BOLD_NL-far_ – BOLD_HPDL_ Thus, positive values reflect the suppression of BOLD responses at the HPDL comparted to the NL-far location. The BOLD suppression index was computed for each stimulus type separately, as in the main ROI analysis (i.e. for Targets, Neutrals and Distractors). Finally, we correlated RT_facilitation_ with BOLD_suppression_ across participants using Pearson correlation. Results showed a small, but not statistically significant correlation between RT facilitation and BOLD suppression for distractor (r_(26)_ = 0.22, p = 0.257), target (r_(26)_ = 0.10, p = 0.598) and neutral (r_(26)_ = 0.13, p = 0.519) stimuli. Thus, while the direc on of the correlation was in line with the specula on by the reviewer in the “ Recommendations for the authors”, results were not statistically reliable and therefore inconclusive. As also noted in our preliminary reply to the reviewer comments, it was a priori unlikely that this analysis would yield a statistically significant correlation. An a priori power analysis suggested that, to reach a power of 0.8 at a standard alpha of 0.05, given the present sample size of n=28, the effect size would need to exceed r > 0.75, which seemed unlikely for the correlation of behavioural and neural difference scores. Given the inconclusive nature of the results, we prefer to not include this additional analysis in the manuscript, as we believe that it does not add to the main message of the paper but have it accessible to the interested reader in the public “peer review process”.

The study provides an advance over previous studies, which iden fied enhancement or suppression in visual cortex as a function of search target/distractor predictability, but in less spatially-specific way. It also speaks to open questions about whether such suppression/enhancement is observed only in response to the arrival of visual information, or instead is preparatory, favouring the la er view. The theoretical advance is moderate, in that it is largely congruent with previous frameworks, rather than strongly excluding an opposing view or providing a major step change in our understanding of how distractor suppression unfolds.

We agree with the reviewer that our results are an advancement of prior work, particularly with respect to narrowing down the role of sensory areas and the proactive nature of distractor suppression. However, we argue that this represents a significant step forward for several reasons. First, to our knowledge, the literature on distractor suppression, and visual search in general, is by no means unanimous with respect to the conclusion that distractor suppression is instantiated proactively (Huang et al., 2021, 2022). Indeed, there are several studies suggesting the opposite account; reactive suppression (Chang et al., 2023) or contributions by both proactive and reactive mechanisms (Sauter et al., 2021; Wang et al., 2019). Moreover, studies in support of proactive distractor suppression did not investigate the involvement of (early) sensory areas during suppression. Conversely, to our knowledge most studies investigating the involvement of sensory cortex during distractor suppression did not address the question whether suppression arises proactive or reactively.

**Recommendations for the authors:**

**Reviewer #1 (Recommendations for the authors):**
Minor Points:(1) There are several disconnects between the behaviour and the MR results - i.e. not stimulus specific yet there are no deficits for targets appearing the HPDL, also no behavioural suppression for the NLNear but neural suppression found. Nevertheless, the behaviour is used as a way to rule out potential attentional strategies when considering whether there is enhancement in the NL-Far condition. I realise you have a few other points here, but I think it's worth addressing what could be seen as a double standard.

The reviewer points out an important concern, which we feel could have better been addressed in the manuscript. From our point of view a partial dissociation between neural modulations in EVC and eventual behavioural facilitation is not surprising, given the extensive neural processing beyond EVC required for behaviour. However, this assessment may differ, if one stresses an explicit volitional attentional strategy over an implicit statistical learning account. That said, we clearly do not want to create the impression of using a double standard. The lack of behavioural facilitation for targets at NLfar is not a critical part of our argument against explicit attentional strategies. Therefore, we rephrased the relevant paragraph in the Discussion section to now emphasize the importance of the control analysis excluding participants who reported the correct HPDL in the questionnaire (Figure 5), but nonetheless yielded qualitatively identical results to the main ROI analysis (Figure 4). In our opinion, this control analysis provides more compelling evidence against a volitional attentional strategy account without the risk of crea ng the impression of applying a double standard in the interpretation of behavioural data. Additionally, we now acknowledge the limitation of relying on behavioral data in ruling out volitional attentional strategies in the updated manuscript:

“It is well established that attention enhances BOLD responses in visual cortex (Maunsell, 2015; Reynolds & Chelazzi, 2004; Williford & Maunsell, 2006). If participants learned the underlying distractor contingencies, they could deploy an explicit strategy by directing their attention away from the HPDL, for example by focusing attention on the diagonally opposite neutral location. This account provides an alternative explanation for the observed EVC modulations. However, while credible, the current findings are not consistent with such an interpretation. First, there was no behavioral facilitation for target stimuli presented at the far neutral location, contrary to what one might expect if participants employed an explicit strategy. However, given the partial dissociation between neural suppression in EVC and behavioral facilitation, additional neural data analyses are required to rule out volitional attention strategies. Thus, we performed a control analysis that excluded all participants that indicated the correct HPDL location in the questionnaire, thereby possibly expressing explicit awareness of the contingencies. This control analysis yielded qualitatively identical results to the full sample, showing significant distractor suppression in EVC. Therefore, it is unlikely that explicit attentional strategies, and the enhancement of locations far from the HPDL, drive the results observed here. Instead the current finding are consistent with an account emphasizing the automa c deployment of spatial priors (He et al., 2022) based on implicitly learned statistical regularities.”

(2) Does the level of suppression change in any way through the experiment? I.e., does it get stronger in the second vs. first half of the experiment?

The reviewer askes an interesting question, whether BOLD suppression may change across the experiment. To address this question, we performed an additional analysis testing BOLD suppression in EVC during the first compared to second half of the MRI experiment. Here we defined BOLD suppression as: BOLD_suppression_ = ((BOLD_NL-far_ – BOLD_HPDL_) + (BOLD_NL-far_ – BOLD_NL-near_)) / 2. Thus, in this formula on of BOLD suppression we summarize the two primary BOLD suppression effects observed in our main results (Figure 4). Additionally, as we previously did not observe any significant differences in BOLD suppression magnitudes between different stimulus types (i.e. suppression was similar for target, distractor and neutral stimuli), we collapsed across stimulus types in this analysis.

Results, depicted below, showed that during both the initial (Run 1+2) and later part (Run 4+5) of the MRI experiment BOLD suppression was statistically significant (BOLD suppression Run 1+2: W = 331, p = 0.003, r = 0.63; BOLD suppression Run 4+5: W = 320, p = 0.007, r = 0.58) , confirming our main results of reliable distractor suppression even in this subset of trials. However, we did not observe any statistically significant differences between early and late runs of the experiment (t_(27)_ = -0.21, p = 0.835, d = -0.04). In fact, a Bayesian paired t-test provided evidence for the absence of a difference in BOLD suppression between early compared to later runs (BF_10_ = 0.205), suggesting that distractor suppression in EVC was stable throughout the experiment. A qualitatively similar, pattern was evident during omission trials, with significant distractor suppression during early runs (t_(27)_ = 2.70, p = 0.012, d = 0.51), but not quite a statistically significant modulation for later runs (t_(27)_ = 1.97, p = 0.059, d = 0.37). Again, there was no evidence for a difference in suppression magnitudes across the experiment (W = 198, p = 0.920, d = -0.025) and support for the absence of a difference in BOLD suppression between early and late runs (BF_10_ = 0.278).

**Author response image 1. sa3fig1:** Analysis of BOLD suppression magnitudes in EVC across the MRI experiment phases. BOLD suppression was comparable between early (Run 1+2) and late (Run 4+5) phases of the MRI experiment, suggesting consistent suppression in EVC following statistical learning. Error-bars denote within-subject SEM. * p < 0.05, ** p < 0.01, = BF_10_ < 1/3.

In sum, results suggest that distractor suppression in EVC was stable across runs and did not change significantly throughout the experiment. This result was a priori likely, given that participants already underwent behavioral training before entering the MRI. This enabled them to establish modified spatial priority maps, containing the high probability distractor location contingencies, already before the first MRI run. While specula ve, it is possible that participants may still have consolidated the spatial priority maps during the initial runs, but that this additional consolation is not evident in the data, as later runs may see less engagement by participants due to increasing fa gue towards the end of the MRI experiment. Indeed, rapid learning and stable suppression throughout the remainder of the experiment is also reported by prior work (Lin et al., 2021). We believe that it is highly interesting for future studies to investigate the development of distractor suppression across learning, with initial exposure to the contingencies inside the MRI. However, as the present results are inconclusive, we prefer to not include this analysis in the main manuscript, as it may not provide significant additional insight into the neural mechanisms underlying distractor suppression.

(3) In the methods vs. results you have reported the probabili es slightly differently. In the methods you say the HPDL was 6x more likely to contain a distractor whereas in the results you say 4x. Based on the reported trial numbers I think it should be 4, but probably you want to double check that this is consistent and correct throughout.

We thank the reviewer for bringing this inconsistency to our attention. We have corrected this oversight in the adjusted manuscript:

“One of the four locations of interest was designated the high probability distractor location (HPDL), which contained distractor stimuli (unique color) four mes more o en than any of the remaining three locations of interest. In other words, if a distractor was present on a given trial (42 trials per run), the distractor appeared 57% (24 trials per run) at the HPDL and at one of the other three locations with equal probability (i.e., 14% or 6 trials per run per location).”

**Reviewer #2 (Recommendations for the authors):**
The authors have performed their analyses in the volume rather than the surface, and have grouped together V1, V2, and V3 as "early visual cortex". As the authors' claims lean heavily on the idea that they are measuring "early" visual responses, the study would be improved by delinea ng the ROIS within these different retinotopic regions. Such an approach might be facilitated by analysing data on the reconstructed surface.

Please refer to our reply to this analysis suggested in the Public review.

The authors rightly tread carefully on the causal link between their neural findings and the behavioural outcomes. The picture might be clarified somewhat further by testing for a positive relationship between behavioural effect sizes and neural effect sizes across participants. e.g. to what extent is the search advantage when distractors are presented at the "HPDL" linked to greater suppression of BOLD at the HDPL region of early visual cortex?

Please refer to our reply to this analysis suggested in the Public review.

Some of the claims based on null hypotheses would be better supported by Bayesian tests e.g. page 6 "This pattern of results was the same regardless whether the distractor, target, or a neutral stimulus presented at the HPDL and NL-near locations compared to NL-far ..." and "BOLD responses between HPDL and NL-near locations did not reliably differ ..." This is similar to the approach that the authors adopted later in the section "Ruling out attentional modulation".

We agree with the reviewer that our ROI analyses would benefit from providing evidence for the absence of a modulation. Accordingly, we updated our results by adding equivalent Bayesian tests. Bayes Factors were computed using JASP 0.18.2 (JASP Team, 2024; RRID:SCR_015823) with default settings; i.e. for Bayesian paired t-tests with a Cauchy prior width of 0.707. Qualitative interpretations of BFs were based on Lee and Wagenmakers (2014). We now report the obtained BF in the Results section.

“BOLD responses between HPDL and NL-near locations did not reliably differ (HPDL vs NL-near: t_(27)_ = 0.47, p_holm_ = 0.643, d = 0.08; BF_10_ = 0.19).”

And:

“Neural responses at HPDL and NL-near did not reliably differ (t_(27)_ = 0.21, p_holm_ = 0.835 d = 0.04; BF_10_ = 0.21).”

Moreover, we now denote any equivalent results (defined as BF_10_<1/3) in Fig. 4 and Fig. 5, and included the descrip on of the associated symbol in the figure text (“ = BF_10_ < 1/3”).

Additionally, we now also report the BF for all paired t-tests reported in Supplementary Table 1.

Finally, we addressed the statement: “This pattern of results was the same regardless whether the distractor, target, or a neutral stimulus presented at the HPDL and NL-near locations compared to NLfar”. Our inten on was to emphasize that the pattern of results reported in the sentence preceding it was evident for distractor, target, or neutral stimulus, and not to suggest that the magnitude of the effect is the same. Hence, to more accurate reflect the results, we changed this sentence to: “This pattern of results was present regardless whether the distractor, target, or a neutral stimulus presented at the HPDL and NL-near locations compared to NL-far”